# MULTI-FIELD ADAPTIVE RETRIEVAL

**Millicent Li**[1‡], **Tongfei Chen**[2‡], **Benjamin Van Durme**[3], **Patrick Xia**[3]
[1]Northeastern University, [2]Augment Code, [3]Microsoft
li.mil@northeastern.edu, patrickxia@microsoft.com

## ABSTRACT

Document retrieval for tasks such as search and retrieval-augmented generation typically involves datasets that are *unstructured*: free-form text without explicit internal structure in each document. However, documents can have some *structure*, containing fields such as an article title, a message body, or an HTML header. To address this gap, we introduce *Multi-Field Adaptive Retrieval* (MFAR), a flexible framework that accommodates any number and any type of document indices on *semi-structured* data. Our framework consists of two main steps: (1) the decomposition of an existing document into fields, each indexed independently through dense and lexical methods, and (2) learning a model which adaptively predicts the importance of a field by conditioning on the document query, allowing on-the-fly weighting of the most likely field(s). We find that our approach allows for the optimized use of dense versus lexical representations across field types, significantly improves in document ranking over a number of existing retrievers, and achieves state-of-the-art performance for multi-field semi-structured data.

## 1 INTRODUCTION

The task of document retrieval has many traditional applications, like web search or question answering, but there has also been renewed interest as part of LLM workflows, like retrieval-augmented generation (RAG). An area of study is focused on increasing the complexity and naturalness of queries (Yang et al., 2018; Qi et al., 2019; Jeong et al., 2024; Lin et al., 2023). Another less studied area considers the increased complexity of the documents (Jiang et al., 2024; Wu et al., 2024b). This represents a challenge compared to prior datasets for retrieval, like MS MARCO (Nguyen et al., 2016), which contain chunks of text that are highly related to the query. Retrieval is done by either searching over the documents via lexical match (Robertson et al., 1994) or with dense retrievers that embed text into vector representations (Karpukhin et al., 2020; Ni et al., 2022; Izacard et al., 2022). Relatedly, some approaches (Gao et al., 2021; Chen et al., 2022) explore the benefits of a hybrid solution, but these options are not mainstream. In this work, we revisit both hybrid models and methods for retrieval of more complex documents.

Our motivation for this direction derives from two observations: 1) documents do have structure: *fields* like titles, timestamps, headers, authors, etc. and queries can refer directly to this structure; and 2) a different scoring method may be beneficial for each of these fields, as not every field is necessary to answer each query. More specifically, our goal is to investigate retrieval on **semi-structured** data. Existing work on retrieval for semi-structured data with dense representations focus on directly embedding semi-structured knowledge into the model through pretraining approaches (Li et al., 2023; Su et al., 2024), but we would like a method which can more flexibly combine existing pretrained models and scorers. Similarly, there has been prior interest in multi-field retrieval, although these works focused on retrieval with solely lexical or sparse features or early neural models (Robertson et al., 1994; Zaragoza et al., 2004; Zamani et al., 2018).

In this work, we demonstrate how multi-field documents can be represented through paired views and on a per-field basis, with a learned mechanism that maps queries to weighted combinations of these views. Our method, **M**ulti-**F**ield **A**daptive **R**etrieval (MFAR),[1] is a retrieval approach that can accommodate any number of fields and any number of scorers (such as one lexical and one

---

[‡] Work done while at Microsoft

[1] https://github.com/microsoft/multifield-adaptive-retrieval

| Dataset | Example Query | Example Document |
|---|---|---|
| MS MARCO | *aleve maximum dose* | You should take one tablet every 8 to 10 hours until symptoms abate, … |
| BioASQ | *What is Piebaldism?* | Piebaldism is a rare autosomal dominant disorder of melanocyte development characterized by a congenital white forelock and multiple … |
| STaRK-Amazon | *Looking for a chess strategy guide from **The House of Staunton** that offers tactics against **Old Indian and Modern defenses**. Any recommendations?* | Title: Beating the King's Indian and Benoni Defense with 5. Bd3
Brand: **The House of Staunton**
Description: ... This book also tells you how to play against the **Old Indian and Modern defenses**.
Reviews: [{reviewerID: 1234, text:...}, {reviewerID: 1235, text:...}, ...]
… |
| STaRK-MAG | *Does any research from the **Indian Maritime University** touch upon Fe II **energy level transitions** within the scope of **Configuration Interaction**?* | Title: Radiative **transition rates** for the forbidden lines in **Fe II**
Abstract: We report electric quadrupole and magnetic dipole transitions among the levels belonging to 3d 6 4s, 3d 7 and 3d 5 4s 2 configurations of **Fe II** in a large scale **configuration interaction** (CI) calculation. ...
Authors: N.C. Deb, A Hibbert (**Indian Maritime University**)
… |
| STaRK-Prime | *What **drugs** target the **CYP3A4** enzyme and are used to treat **strongyloidiasis**?* | Name: Ivermectin
Entity Type: **drug**
Details: {Description: Ivermectin is a broad-spectrum anti-parasite medication. It was first marketed under..., Half Life: 16 hours}
Target: gene/protein
Indication: For the treatment of intestinal **strongyloidiasis** due to ...
Category: [Cytochrome P-450 **CYP3A Inducers**, Lactones, ...]
… |

Figure 1: Traditional documents for retrieval (top), like in MS MARCO (Nguyen et al., 2016) and BioASQ (Nentidis et al., 2023), are *unstructured*: free-form text that tends to directly answer the queries. Documents in the STaRK datasets (bottom) (Wu et al., 2024b), are *semi-structured*: each contains multiple fields. The queries require information from some of these fields, so it is important to both aggregate evidence across multiple fields while ignoring irrelevant ones.

vector-based) for each field. Additionally, we introduce a light-weight component that adaptively weights the most likely fields, conditioned on the query. This allows us to exhaustively include all fields and scorers at inference and let the model determine the relative importance. MFAR obtains significant performance gains over existing state-of-the-art baselines. Unlike prior work, our simple approach does not require pretraining and offers some controllability at test-time. Concretely, our contributions are:

1. We introduce a novel framework for document retrieval, MFAR, that is aimed at semi-structured data with any number of fields. Notably, MFAR is able to mix lexical and vector-based scorers between the query and the document's fields.

2. We find that a hybrid mixture of scorers performs better than using dense or lexical-based scorers alone; we also find that encoding documents with our multi-field approach can result in better performance than encoding the entire document as a whole. As a result, MFAR achieves state-of-the-art performance on STaRK, a dataset for semi-structured document retrieval.

3. We introduce an adaptive weighting technique that conditions on the query, weighting more the fields most related to the query and weighting less the fields that are less important.

4. Finally, we analyze the performance of our models trained from our framework; we control the availability of scorers at test-time in an ablation study to measure the importance of the individual fields in the corpus.

## 2 MULTI-FIELD RETRIEVAL

While semi-structured documents is a broad term more generally, in this work, we focus on documents that can be decomposed into *fields*, where each field has a name and a value. As an example in Figure 1, for the STaRK-Prime document, Entity Type would be a field name and its value would be "drug." The values themselves can have additional nested structure, like Category has a list of terms as its value. Note that this formulation of semi-structured multi-field document is broad, as it not only includes objects like knowledge base entries, but also free-form text (chat messages, emails) along with their associated metadata (timestamps, sender, etc) and tabular data.

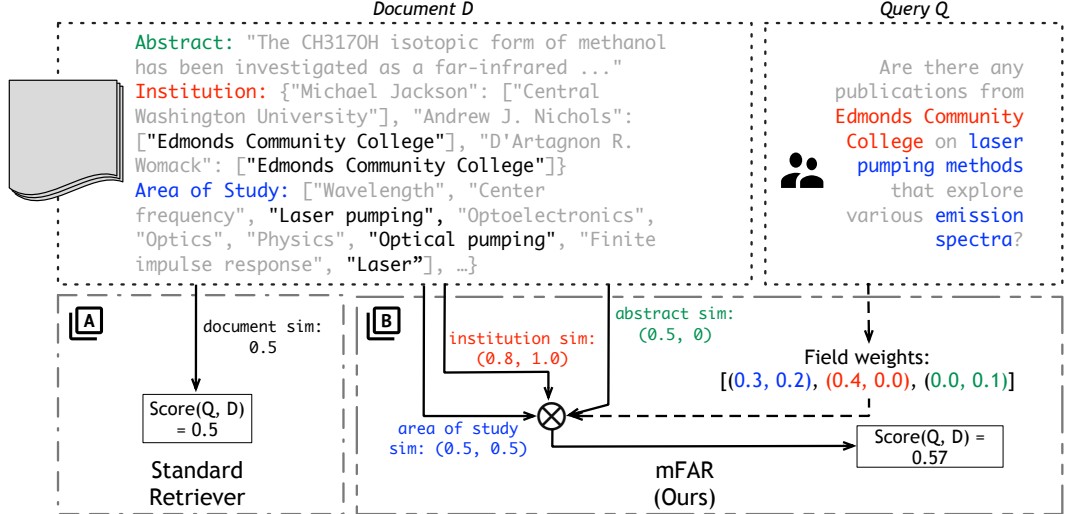

Figure 2: Document $D$ and query $Q$ are examples from the STaRK-MAG dataset. Parts of the query (highlighted) correspond with specific fields from $D$. Traditional retrievers (A) would score the entire document against the query (e.g. through vector similarity). In (B), our method, mFAR, first decomposes $D$ into fields and scores each field separately against the query using both lexical- and vector-based scorers. This yields a pair of field-specific similarity scores, which are combined using our adaptive query conditioning approach to produce a document-level similarity score.

Formally, we consider a corpus of documents $\mathcal{C} = \{d_1, d_2, \ldots, d_n\}$ and a set of associated fields $\mathcal{F} = \{f_1, f_2, \ldots, f_m\}$ that make up each document $d$, i.e., $d = \{f : x_f \mid f \in \mathcal{F}\}$, where $x_f$ is the value for that field. Then, given a natural-language query $q$, we would like a scoring function $s(q, d)$ that can be used to rank the documents in $\mathcal{C}$ such that the most relevant documents to $q$ score highest (or within the top-$k$). $q$ may ask about values from any subset of fields, either lexically or semantically.

## 2.1 Standard Retriever and Contrastive Loss

Traditionally, $d$ is indexed in its entirety. The retriever can employ either a lexical (Robertson et al., 1994) or dense (embedding-based) (Lee et al., 2019; Karpukhin et al., 2020) scorer. A lexical scorer like BM25 (Robertson et al., 1994) directly computes $s(q, d)$ based on term frequencies. For a dense scorer, document and query encoders are used to embed $d$ and $q$, and a simple similarity function, in our case an unnormalized dot product, is used to compute $s(q, d)$.

Document and query encoders can be finetuned by using a contrastive loss (Izacard et al., 2022), which aims to separate a positive (relevant) document $d_i^+$ against $k$ negative (irrelevant) documents $\mathcal{D}_i^-$ for a given query $q$. In prior work, a shared encoder for the documents and queries is trained using this loss, and a temperature $\tau$ is used for training stability:

$$\mathcal{L}_c = -\log \frac{e^{s(q_i, d_i^+)/\tau}}{e^{s(q_i, d_i^+)/\tau} + \sum_{d_i^- \in \mathcal{D}_i^-} e^{s(q_i, d_i^-)/\tau}} \tag{1}$$

$\mathcal{L}_c$ is the basic contrastive loss which maximizes $P(d_i^+ \mid q_i)$. Following Henderson et al. (2017); Izacard et al. (2022) and Chen et al. (2020a), we employ in-batch negatives to efficiently sample those negative documents by treating the other positive documents in the batch (of $b$ documents), $d_j^+$, as negatives and including them into $\mathcal{D}_i^-$, where, $j \neq i$ and $1 \leq j \leq b$. Furthermore, following prior work (Yang et al., 2019; Ni et al., 2022; Chen et al., 2025), we can include a bi-directional loss for $P(q_i \mid d_i^+)$. Here, for a given positive document $d_i^+$, $q_j$ is the positive *query* and the other queries $q_i, i \neq j$ become negative queries:

$$\mathcal{L}_b = -\log \frac{e^{s(q_i, d_i^+)/\tau}}{e^{s(q_i, d_i^+)/\tau} + \sum_{q_j, j \neq i} e^{s(q_j, d_i^+)/\tau}} \tag{2}$$

The final loss for the (shared) encoder is $\mathcal{L} = \mathcal{L}_c + \mathcal{L}_b$.

## 2.2 MFAR: A MULTI-FIELD ADAPTIVE RETRIEVER

Because semi-structured documents can be decomposed into individual fields ($d = \{x_f\}_{f \in \mathcal{F}}$), we can score the query $q$ against each field separately. This score could be computed via **lexical** or **dense** (vector-based) methods. This motivates a modification to the standard setup above, where $s(q, d)$ can instead be determined as a weighted combination of field-wise scores and scoring methods,

$$s(q, d) = \sum_{f \in \mathcal{F}} \sum_{m \in \mathcal{M}} w_f^m s_f^m(q, x_f). \tag{3}$$

Here, $s_f^m(q, x_f)$ is the score between $q$ and field $f$ of $d$ using scoring method $m \in \mathcal{M}$, and $\mathcal{M}$ is the set of scoring methods. For a hybrid model, $\mathcal{M} = \{\text{lexical}, \text{dense}\}$. $w_f^m$ is a weight, possibly learned, that is associated with field $f$ and scoring method $m$.

**Adaptive field selection.** As presented, our method uses weights, $w_f^m$, that are learned for each field and scorer. This is useful in practice, as not every field in the corpus is useful or even asked about, like unrelated numbers or internal identifiers. Additionally, queries usually ask about information contained in a small number of fields **and** these fields change depending on the query.

This motivates *conditioning* the value of $w_f^m$ also on $q$ so that the weights can adapt to the given query by using the query text to determine the most important fields. We use an adaptation function $G$ and let $w_f^m = G(q, f, m)$. Now, the query-conditioned, or adaptive, weighted sum is:

$$s(q, d) = \sum_{f \in \mathcal{F}} \sum_{m \in \mathcal{M}} G(q, f, m) \cdot s_f^m(q, x_f). \tag{4}$$

To implement $G$, let $\mathbf{q}$ be a dense embedding of $q$, and $\mathbf{a}_f^m \in \mathbb{R}^{|\mathbf{q}|}$ be learnable parameters. Then we could define $G(q, f, m) = \mathbf{a}_f^{m\top}\mathbf{q}$. We find that learning is more stable with a nonlinearity over all fields $f$ and scorers $m$: $G(q, f, m) = \text{softmax}(\{\mathbf{a}_f^{m\top}\mathbf{q}\})$, which is what we use in MFAR.

**Multiple scorers and normalization.** One objective of ours is to seamlessly incorporate scorers using different methods (lexical and dense). However, the distribution of possible scores per scorer can be on different scales. While $G$ can technically learn to normalize, we want $G$ to focus on query-conditioning. Instead, we experiment with using batch normalization (Ioffe & Szegedy, 2015) per field that whitens the scores and learns new scalars $\gamma_f^m$ and $\beta_f^m$ for each field and scorer. Because these scores are ultimately used in the softmax of the contrastive loss, $\gamma_f^m$ acts like a bias term which modulates the importance of each score while $\beta_f^m$ has no effect.

Note that the score whitening process is not obviously beneficial or necessary, especially if the scorers already share a similar distribution (i.e. if we only use dense scorers). We leave the inclusion of normalization as a hyperparameter as part of our grid search.

**Inference** At test time, the goal is to rank documents by $s(q, d)$ such that the relevant (gold) documents are highest. Because it can be slow to compute $|\mathcal{F}||\mathcal{M}||\mathcal{C}|$ scores for the whole corpus, we use an approximation. We first determine a top-$k$ shortlist, $\mathcal{C}_f^m$, of documents for each field and scorer and only compute the full scores for all $\bigcup_{f \in \mathcal{F}, m \in \mathcal{M}} \mathcal{C}_f^m$, which results in the final ranking. Note this inexact approximation of the top-$k$ document is distinct from traditional late-stage re-ranking methods that rescore the query with each document, which is not the focus of this work.

## 3 EXPERIMENTS

Our experiments are motivated by the following hypotheses:

1. Taking advantage of the multi-field document structure will lead to better accuracy than treating the document in its entirely, as a single field.
2. Hybrid (a combination of lexical and dense) approaches to modeling will perform better than using only one or other.

## 3.1 DATA

We use STaRK (Wu et al., 2024b), a collection of three retrieval datasets in the domains of product reviews (Amazon), academic articles (MAG), and biomedical knowledge (Prime), each derived from knowledge graphs. Amazon contains queries and documents from Amazon Product Reviews (He & McAuley, 2016) and Amazon Question and Answer Data (McAuley et al., 2015). MAG contains queries and documents about academic papers, sourced from the Microsoft Academic Graph (Wang et al., 2020), obgn-MAG, and obgn-papers100M (Hu et al., 2020). Prime contains queries and documents regarding biomedicine from PrimeKG (Chandak et al., 2022). These datasets are formulated as knowledge graphs in STaRK and are accompanied by complex queries.

In the retrieval baselines (Wu et al., 2024b), node (corresponding to an entity in the knowledge graph) information is linearized into documents that can be encoded and retrieved via dense methods. We likewise treat each node as a document. In our work, we preserve each node property or relation as a distinct field for our *multi-field* models or likewise reformat to a human-readable document for our *single-field* models. Compared to Amazon and MAG, we notice that Prime contains a higher number of relation types, i.e. relatively more fields in Prime are derived from knowledge-graph relations than in either Amazon or MAG, where document content is derived from a node's properties. In total, there are 22, 8, and 5 fields for Prime, Amazon, and MAG respectively; more details on dataset sizes, preprocessing, exact fields are described in Appendix A.

We use `trec_eval`[2] for evaluation and follow Wu et al. (2024b) by reporting Hit@1, Recall@20, and mean reciprocal rank (MRR). Hit@5 is reported in Appendix C.1 due to space limitations here.

## 3.2 BASELINES AND PRIOR WORK

We compare primarily to prior work on STaRK, which is a set of baselines established by Wu et al. (2024b) and more recent work by Wu et al. (2024a). Specifically, they include two vector similarity search methods that use OpenAI's `text-embedding-ada-002` model, *ada-002* and *multi-ada-002*. Notably, the latter is also a multi-vector approach, although it only uses two vectors per document: one to capture node properties and one for relational information. We also include their two LLM-based re-ranking baselines (Claude3 and GPT4 rerankers) on top of ada-002. Although our work does not perform re-ranking, we add these results to show the superiority of finetuning smaller retrievers over using generalist LLMs for reranking.

More recently, AvaTaR (Wu et al., 2024a) is an agent-based method which iteratively generates prompts to improve reasoning and scoring of documents. While not comparable with our work, which does not focus on agents nor use models as large, it is the state-of-the-art method for STaRK.

Finally, we use an off-the-shelf pretrained retrieval encoder, Contriever finetuned on MS MARCO[3](Izacard et al., 2022), as a baseline for our dense scorer, which we subsequently continue finetuning on STaRK. Early experiments showed that Contriever performed better than other dense retrievers. We use BM25 (Robertson et al., 1994; Lù, 2024) as a lexical baseline. These use the single-field formatting described Section 3.1.

## 3.3 EXPERIMENTAL SETUP

MFAR affords a combination of lexical and dense scorers across experiments. Similarly to our baselines, we use BM25 as our lexical scorer and the dot product of Contriever embeddings as our dense scorer. We use a shared embedding model for both the query and document and for creating $\mathbf{q}$ when computing the adaptation function $G$. Because of potential differences across datasets, we initially consider four configurations that take advantage of MFAR's ability to accommodate multiple fields or scorers: $\text{MFAR}_{\text{Dense}}$ uses all fields and the dense scorer, $\text{MFAR}_{\text{Lexical}}$ uses all fields and the lexical scorer, $\text{MFAR}_{\text{All}}$ uses all fields and both scorers, and $\text{MFAR}_2$ uses both scorers but the single-field (Sec. 3.1) document representation. Based on our final results and analysis, we additionally create and evaluate $\text{MFAR}_{\text{All+2}}$, which consists of both a single-document and multi-field representation for both lexical and dense scoring methods. This results in five MFAR models that use $|\mathcal{F}|$, $|\mathcal{F}|$, $2|\mathcal{F}|$, 2, and $2|\mathcal{F}| + 2$ scorers respectively. For each dataset (and across models),

---

[2] `https://github.com/usnistgov/trec_eval`.
[3] `https://huggingface.co/facebook/contriever-msmarco`.

Table 1: Comparing our method (MFAR) against baselines and state-of-the-art methods on the STaRK test sets. ada-002 and multi-ada-002 are based on vector similarity; +{Claude3, GPT4} further adds an LLM reranking step on top of ada-002. AvaTaR is an agent-based iterative framework. Contriever-FT is a finetuned Contriever model, which is also the encoder finetuned in MFAR. MFAR is superior against prior methods and datasets, and earns a substantial margin on average across the benchmark. ⋄ In Wu et al. (2024b), these are reranker models that are only run on a random 10% subset of queries.

| Model | Amazon | | | MAG | | | Prime | | | Average | | |
|---|---|---|---|---|---|---|---|---|---|---|---|---|
| | H@1 | R@20 | MRR | H@1 | R@20 | MRR | H@1 | R@20 | MRR | H@1 | R@20 | MRR |
| ada-002 | 0.392 | 0.533 | 0.542 | 0.291 | 0.484 | 0.386 | 0.126 | 0.360 | 0.214 | 0.270 | 0.459 | 0.381 |
| multi-ada-002 | 0.401 | 0.551 | 0.516 | 0.259 | 0.508 | 0.369 | 0.151 | 0.381 | 0.235 | 0.270 | 0.480 | 0.373 |
| Claude3$^\diamond$ | 0.455 | 0.538 | 0.559 | 0.365 | 0.484 | 0.442 | 0.178 | 0.356 | 0.263 | 0.333 | 0.459 | 0.421 |
| GPT4$^\diamond$ | 0.448 | 0.554 | 0.557 | 0.409 | 0.486 | 0.490 | 0.183 | 0.341 | 0.266 | 0.347 | 0.460 | 0.465 |
| AvaTaR agent | 0.499 | 0.606 | 0.587 | 0.444 | 0.506 | 0.512 | 0.184 | 0.393 | 0.267 | 0.376 | 0.502 | 0.455 |
| BM25 | 0.483 | 0.584 | 0.589 | 0.471 | 0.689 | 0.572 | 0.167 | 0.410 | 0.255 | 0.374 | 0.561 | 0.462 |
| Contriever-FT | 0.383 | 0.530 | 0.497 | 0.371 | 0.578 | 0.475 | 0.325 | 0.600 | 0.427 | 0.360 | 0.569 | 0.467 |
| MFAR$_{\text{Lexical}}$ | 0.332 | 0.491 | 0.443 | 0.429 | 0.657 | 0.522 | 0.257 | 0.500 | 0.347 | 0.339 | 0.549 | 0.437 |
| MFAR$_{\text{Dense}}$ | 0.390 | 0.555 | 0.512 | 0.467 | 0.669 | 0.564 | 0.375 | 0.698 | 0.485 | 0.411 | 0.641 | 0.520 |
| MFAR$_2$ | **0.574** | **0.663** | **0.681** | 0.503 | 0.721 | 0.603 | 0.227 | 0.495 | 0.327 | 0.435 | 0.626 | 0.537 |
| MFAR$_{\text{All}}$ | 0.412 | 0.585 | 0.542 | 0.490 | 0.717 | 0.582 | **0.409** | 0.683 | **0.512** | 0.437 | 0.662 | 0.545 |
| MFAR$_{\text{All+2}}$ | 0.530 | **0.663** | 0.643 | **0.559** | **0.741** | **0.643** | 0.400 | **0.726** | 0.520 | **0.496** | **0.710** | **0.602** |

we run a grid search over learning rates and whether to normalize and select the best model based on the development set.

Because Contriever has a 512-token context window, we prioritize maximizing this window size for each field, which ultimately reduces the batch size we can select for each dataset, resulting in 96 for Amazon and Prime, and 192 for MAG. More details on the exact hyperparameters for each run are in Appendix B.

## 4 RESULTS

We report the results from our MFAR models in Table 1, compared against against prior methods and baselines. Our best models—both make use of both scorers—perform significantly better than prior work and baselines: MFAR$_2$ on Amazon, and MFAR$_{\text{All}}$ and MFAR$_{\text{All+2}}$ on the other datasets. This includes surpassing re-ranking based methods and the strongest agentic method, AvaTaR. MFAR$_{\text{All}}$ performs particularly well on Prime (+20% for H@1). Comparatively, all models based on ada-002 have extended context windows of 2K tokens, but MFAR, using an encoder that has a much smaller context window size (512), still performs significantly better. Furthermore, our gains cannot be only attributed to finetuning or full reliance on lexical scorers since the MFAR models perform better against the already competitive BM25 and finetuned Contriever baselines.

We find that the adoption of a hybrid approach benefits recall, which we can attribute to successful integration of BM25's scores. Individually, BM25 already achieves higher R@20 than most vector-based methods. The MFAR models retain and further improve on that performance. Recall is especially salient for tasks such as RAG where collecting documents in the top-$k$ are more important than surfacing the correct result at the top.

Revisiting our hypotheses from Section 3, we can compare the various configurations of MFAR. Noting that BM25 is akin to a single-field, lexical baseline and Contriever-FT is a single-field, dense baseline, we can observe the following:

**Multi-field vs. Single-field.** A side-by-side comparison of the single-field models against their multi-field counterparts shows mixed results. If we only consider dense scorers, MFAR$_{\text{Dense}}$ produces the better results than Contriever-FT across all datasets. To our knowledge, this is the first positive evidence in favor of multi-field methods in dense retrieval. For MFAR$_{\text{Lexical}}$, in both Amazon and MAG, the BM25 baseline performs especially well, and we do not see consistent improvements. This specific phenomenon has been previously noted by Robertson et al. (2004), who describe BM25F, a variant of BM25 that aggregates multi-field information in a more principled manner.

Specifically in BM25, the scores are length normalized. For some fields, like institution, repetition does not imply a stronger match, and so treating the institution field separately (and predicting high weights for it) could lead to high scores for negative documents. A multi-field sparse representation, then, may not always be the best solution, depending on the dataset. We further try using BM25F instead but find lackluster performance likely due to undertuned weights (Appendix D). Given the number of fields in STaRK datasets, tuning field weights is a challenging and open problem that appears less tractable for BM25F than for MFAR. Finally, we note that combining multi-field with single-field can lead to further gains, as demonstrated by MFAR$_{\text{All+2}}$ (and Appendix C.2).

**Hybrid is best.** Across both multi-field (MFAR$_{\text{All}}$ vs. MFAR$_{\text{Dense}}$ or MFAR$_{\text{Lexical}}$) and single-field models (MFAR$_2$ vs. BM25 or Contriever-FT), and across almost every dataset, there is an increase in performance when using both scorers over a single scorer type, validating our earlier hypothesis. This reinforces findings from prior work (Gao et al., 2021; Kuzi et al., 2020) that hybrid methods work well. The one exception (Prime, single-field) may be challenging for single-field models, possibly due to the relatively higher number of fields in the dataset and the semantics of the fields, as we investigate more in Section 5.3. However, in the multi-field setting for Prime, we again see hybrid perform best. This provides evidence for our original motivation: that hybrid models are suitable for and positively benefits certain semi-structured, multi-field documents.

## 5 ANALYSIS

Next, we take a deeper look into why MFAR leads to improvements. We first verify that model is indeed adaptive to the queries by showing that query conditioning is a necessary component of MFAR. Because the field weights are naturally interpretable and controllable, we can manually set the weights to perform a post-hoc analysis of the model, which both shows us which fields of the dataset are important for the given queries and whether the model is benefiting from the dense or lexical scorers, or both, for each field. Finally, we conduct qualitative analysis to posit reasons why MFAR holds an advantage.

Our analyses, along with the quantitative results, lead us to experiment with a combination of single-field and multi-field document representations in Appendix C.2, like MFAR$_{\text{All+2}}$. We find that these combinations offer additional gains over MFAR$_{\text{All}}$, and that even just a combination of single-field lexical and multi-field dense improves over only MFAR$_{\text{All}}$ or MFAR$_2$.

### 5.1 IS QUERY-CONDITIONED ADAPTATION NECESSARY?

We designed MFAR with a mechanism for adaptive field selection: for a test-time query, the model makes a weighted prediction over the fields to determine which ones are important. In this section, we analyze whether this adaptation is necessary to achieve good performance. To do so, we MFAR against an ablated version which does not have the ability to predict query-specific weights but can still predict global, field-specific weights by directly learning $w_f^m$ from Equation 3. This allows the model to still emphasize (or de-emphasize) certain fields globally if they are deemed important (or unimportant).

Table 2: The test scores of MFAR$_{\text{All}}$ without query conditioning (QC) and the % relative change without it.

| | Amazon | | | MAG | | | Prime | | | STaRK Avg. | | |
|---|---|---|---|---|---|---|---|---|---|---|---|---|
| | H@1 | R@20 | MRR | H@1 | R@20 | MRR | H@1 | R@20 | MRR | H@1 | R@20 | MRR |
| MFAR$_{\text{All}}$ | 0.412 | 0.585 | 0.542 | 0.490 | 0.717 | 0.582 | 0.409 | 0.683 | 0.512 | 0.437 | 0.662 | 0.545 |
| No QC | 0.346 | 0.547 | 0.473 | 0.428 | 0.662 | 0.528 | 0.241 | 0.596 | 0.368 | 0.338 | 0.602 | 0.456 |
| Loss (%) | -16.0 | -6.5 | -12.7 | -12.7 | -7.7 | -9.3 | -41.1 | -12.7 | -28.1 | -22.6 | -9.1 | -16.3 |

In Table 2, we present the details for MFAR$_{\text{All}}$ and find that query conditioning is indeed necessary for performance gains across all datasets. Omitting it results in substantial losses on the metrics on each dataset and for the STaRK average. This extends to the other models too. We also find lower scores on STaRK average across the 3 metrics (H@1, R@20, MRR): -10%, -6%, -8% for MFAR$_{\text{Dense}}$ and -17%, -13%, -14% for the MFAR$_{\text{Lexical}}$.

## 5.2 WHICH FIELDS AND SCORERS ARE IMPORTANT?

The interpretable design of our MFAR framework enables us to easily control the used fields and scorers after a model has been trained. Specifically, we can mask (zero) out any subset of the weights $w_f^m$ used to compute $s(q, d)$ (Equation 4). For example, setting $w_f^{\text{lexical}} = 0$ for each $f$ would force the model to only use the dense scores for each field. We can interpret a drop in performance as a direct result of excluding certain fields or scorers, and thus we can measure their contribution (or lack thereof). In this deep-dive analysis, we re-evaluate MFAR$_{\text{All}}$'s performance on each dataset after masking out entire scoring methods (lexical or dense), specific fields (title, abstract, etc), and even specific field and scoring method (e.g. title with dense scorer).

**Scorers** We present results on the three STaRK datasets in Table 3. We see the performance of MFAR$_{\text{All}}$ on Amazon is heavily reliant on the dense scores. Knowing the results from Table 1, this may be unsurprising because MFAR$_{\text{Lexical}}$ did perform the worst. While the model leans similarly towards dense scores for Prime, on MAG, it relies more on the lexical scores. This shows that each dataset may benefit from a different scorer. Further, this may not be expected *a priori*: we would have expected Prime to benefit most from the lexical scores, as that biomedical dataset contains many initialisms and IDs that are not clearly semantically meaningful. This demonstrates the flexibility and adaptivity of MFAR to multiple scoring strategies.

From Table 1, we observe that MFAR$_{\text{All}}$ outperforms MFAR$_{\text{Dense}}$ by a small margin (0.435 vs. 0.411 for average H@1), and so one may suspect MFAR$_{\text{All}}$ is heavily relying on the dense scores. However, MFAR$_{\text{All}}$ with $w_f^{\text{lexical}}$ masked out performs substantially worse on each dataset (Table 3; 0.326 average) than MFAR$_{\text{Dense}}$, suggesting that a nontrivial amount of the performance on MFAR$_{\text{All}}$ is attributable to lexical scores. Thus, unlike late-stage reranking or routing models for retrieval, the coexistence of dense and lexical scorers (or even individual fields) during training likely influences what the model and encoder learns.

Table 3: Performance of MFAR$_{\text{All}}$ with entire scoring methods masked out at test-time.

| Masking | Amazon | | | MAG | | | Prime | | |
|---|---|---|---|---|---|---|---|---|---|
| | H@1 | R@20 | MRR | H@1 | R@20 | MRR | H@1 | R@20 | MRR |
| None | 0.412 | 0.586 | 0.542 | 0.490 | 0.717 | 0.582 | 0.409 | 0.683 | 0.512 |
| Dense only: $w_f^{\text{lexical}} = 0$ | 0.389 | 0.553 | 0.512 | 0.257 | 0.481 | 0.355 | 0.331 | 0.635 | 0.352 |
| Lexical only: $w_f^{\text{dense}} = 0$ | 0.271 | 0.452 | 0.386 | 0.352 | 0.602 | 0.446 | 0.267 | 0.500 | 0.442 |

**Fields** By performing similar analysis at a fine-grained field-level, we can identify which parts of the document are asked about or useful. For each field $f_i$, we can set $w_{f_i}^{\text{lexical}} = 0$, $w_{f_i}^{\text{dense}} = 0$, or both. We collect a few interesting fields from each dataset into Table 4, with all fields in Appendix E.

We find that behaviors vary depending on the field. For some fields (MAG's *authors*, Amazon's *title*), masking out one of the scorers results in almost no change. However, masking out the other one results in a sizeable drop of similar magnitude to masking out both scorers for that field. In this case, one interpretation is that $s_{\text{author}}^{\text{dense}}(q, d)$ and $s_{\text{title}}^{\text{lexical}}(q, d)$ are not useful within MFAR$_{\text{All}}$.

To simplify the model, one may suggest removing any $s_f^m(q, d)$ where setting $w_f^m = 0$ results in no drop. However, we cannot do this without hurting the model. In other words, low deltas do not signify low importance. For some fields (e.g. Amazon's *qa*, MAG's *title*, or Prime's *phenotype absent*), when the lexical or dense scorers are zeroed out individually, the scores are largely unaffected. However, completely removing the field by zeroing both types of scorers results in a noticeable drop. In many cases, we observe that masking out entire fields yields a larger drop than masking out either one individually. This type of behavior could be a result of MFAR redundantly obtaining the same similarity information using different scorers. On the contrary, there is also information overlap across fields, and so in some cases, it is possible in some cases to remove entire fields, especially in Prime (e.g. *enzyme*) and Amazon, without substantial drops.

Table 4: For each dataset, the absolute change (delta) of masking out certain fields and scorers from MFAR$_{\text{All}}$ for H@1 and R@20. For each field, we zero out either the lexical scorer, the dense scorer, or both. The raw scores on all metrics for all fields in each dataset are in Appendix E.

| | Field | $w_f^{\text{lexical}} = 0$ | | $w_f^{\text{dense}} = 0$ | | Both | |
|---|---|---|---|---|---|---|---|
| | | H@1 | R@20 | H@1 | R@20 | H@1 | R@20 |
| Amazon | qa | 0 | 0 | 0 | 0 | -0.031 | -0.041 |
| | title | 0.002 | -0.003 | -0.022 | -0.031 | -0.023 | -0.024 |
| MAG | authors | -0.152 | -0.117 | 0 | 0 | -0.101 | -0.086 |
| | title | -0.011 | -0.003 | -0.017 | -0.014 | -0.076 | 0.063 |
| Prime | phenotype absent | -0.001 | -0.002 | 0 | 0 | -0.033 | -0.030 |
| | enzyme | 0 | 0 | 0 | 0 | -0.004 | -0.006 |

*Query: Which **gene or protein** is **not expressed** in **female gonadal tissue**?*

MFAR$_2$:
name: NUDT19P5
type: **gene/protein**
expression present: {anatomy: **female gonad** }

MFAR$_{\text{All}}$:
name: HSP90AB3P
type: **gene/protein**
expression absent: {anatomy: [cerebellum, **female gonad**]}

*Query: Does Arxiv have any **research papers** from **Eckerd College** on the **neutron scattering** of **6He** in Neutron physics?*

MFAR$_{\text{Lexical}}$:
Abstract: Abstract A new pin-hole small-angle **neutron scattering** (SANS) spectrometer, installed at the cold **neutron** source of the 20 MW China Mianyang Research Reactor (CMRR) in the Institute of Nuclear **Physics** ...
Authors: Mei Peng (China Academy of Engineering **Physics**), Guanyun Yan (China Academy of Engineering **Physics**), Qiang Tian (China Academy of Engineering **Physics**), ...

MFAR$_{\text{Dense}}$:
Abstract: Abstract Measurements of **neutron** elastic and inelastic **scattering** cross sections from 54Fe were performed for nine incident **neutron** energies between 2 and 6 MeV ...
Cited Papers: **Neutron scattering** differential cross sections for 23 Na from 1.5 to 4.5 MeV, **Neutron inelastic scattering** on 54Fe
Area of Study: [**Elastic scattering**, **Physics**, **Inelastic scattering**, **Neutron**, Direct coupling, Atomic physics, **Scattering**, ... ]

MFAR$_{\text{All}}$:
Abstract: ...scattering of **6He** from a proton target using a microscopic folding potential, in which the **6He nucleus** is described in terms of a 4He-core with two additional **neutrons** in the valence p-shell. In contrast to the previous work of that nature, all contributions from the interaction of the **valence neutrons** ...
Authors: P. Weppner (**Eckerd College**), A. Orazbayev (Ohio University), Ch. Elster (Ohio University)
Area of Study: [**elastic scattering**, **physics**, **neutron**, ..., **atomic physics**, **scattering**]

Figure 3: Snippets from the highest-scoring document selected by various MFAR. Top: a single-field hybrid model (MFAR$_2$) vs. MFAR$_{\text{All}}$. MFAR$_{\text{All}}$ picks correctly while MFAR$_2$ is possibly confused by negation in the query. Bottom: Snippets from configurations of MFAR with access to different scorers. Only MFAR$_{\text{All}}$ correctly makes use of both lexical and semantic matching across fields.

## 5.3 QUALITATIVE ANALYSIS

**Multi-field gives semantic meaning for a choice of field, as compared to single-field.** In Figure 3 (top), the query is looking for either a gene or protein that is *not expressed*. With MFAR$_{\text{All}}$, the retriever matches a longer text more accurately than MFAR$_2$ does. Both MFAR$_{\text{All}}$ and MFAR$_2$ correctly match *female gonad*. However, MFAR$_{\text{All}}$ selects the field that refers to the absence of an expression, which is learned by the model. In MFAR$_2$, because the lexical scorer cannot distinguish between present and absent, MFAR$_2$ incorrectly ranks the negative document higher.

**Hybrid excels when both lexical matching and semantic similarity is required.** In Figure 3 (bottom), MFAR$_{\text{All}}$ has the advantage over MFAR$_{\text{Dense}}$ by having the ability to lexically match *Eckerd College*. Furthermore, MFAR$_{\text{All}}$ is still able to semantically match the abstract of the document. While MFAR$_{\text{Dense}}$ also finds a close fit, it is unable to distinguish this incorrect but similar example from the correct one.

We likewise observe the drawbacks of a lexical-only scoring. One limitation of BM25 is that the frequency of successive term matching results in increased scores. Because *Physics* is a keyword with high frequency in the authors list, it results in a high score for this document even though it is not used in the same sense semantically. On the other hand, MFAR$_{\text{All}}$ correctly matches the specific institution because the final scores are based on a weighted combination of lexical and dense scorers, which may reduce or the impact of high lexical scores.

# 6 RELATED WORK

**Structured and Semi-structured Retrieval** Forms of structured and semi-structured retrieval have been explored in a variety of tasks and domains. In particular, we focus on multi-field retrieval, a form of semi-structured retrieval (Zaragoza et al., 2004) for which prior sparse approaches include the aforementioned BM25F (Robertson et al., 2004), learned sparse representations (Zamani et al., 2018), and Bayesian approaches (Piwowarski & Gallinari, 2003). Lin et al. (2023) approach a similar task using dense retrievers and with a primary focus on query decomposition to support a weighted combination of "expert" retrievers. In contrast to their system in which weights are hand-picked and each retriever is independently trained, our MFAR model is learned end-to-end with a single shared encoder and learned weights, which aids scalability.

Table retrieval (Zhang & Balog, 2021; Bhagavatula et al., 2015; Pasupat & Liang, 2015; Herzig et al., 2021; Shraga et al., 2020; Chen et al., 2020b) is a structured retrieval task which adopts similar methods as multi-field retrieval (see Appendix F for an evaluation of MFAR for this task). Table retrieval does not necessarily require table-specific model design, as linearized forms of the table can be adequate for competitive performance (Wang et al., 2022) and many table retrieval datasets have been seen during encoder pretraining (e.g. DPR (Karpukhin et al., 2020) has been trained on Wikipedia). Beyond tabular data, other structured retrieval tasks include code search (Husain et al., 2020) and knowledge graph datasets like shopping (Reddy et al., 2022). The latter is similar to the Amazon subset of STaRK. Separately, a books QA dataset used by Lin et al. (2023) is multimodal, which is not our focus.

Besides decomposition, which targets parts of the structure, like fields, with specialized parameters, prior work has also investigated modifying the training process through generating pseudo-queries based on Wikipedia formatting (Su et al., 2024) and incorporating auxiliary alignment objectives between the document and a natural language description (Li et al., 2023). These methods generally assume that there exist semantically repetitive information (e.g. table and description of table). We do not make this assumption and focus on post-training methods that can use off-the-shelf encoders.

**Hybrid Methods** The combination of both types of lexical and dense scorers has previously been found to be complementary, leading to performance gains (Gao et al., 2021; Kuzi et al., 2020; Lee et al., 2023). Notably, Kuzi et al. (2020) points out that long documents are challenging for lexical-based methods and suggests document chunking as a possible remedy in future work. We implicitly segment the document by taking advantage of the multi-field structure inherently present in documents, and unlike those past works, our work is the first to demonstrate the strength of hybrid-based methods in a multi-field setting. Alternative hybrid retrieval setups combine both dense and lexical features with a dense encoder, trained end-to-end (Lin & Lin, 2023; Shen et al., 2023), whereas we explicitly use existing lexical scorers.

# 7 CONCLUSION

We present MFAR, a novel framework for retrieval over multi-field data by using multiple scorers, each independently scoring the query against a part (field) of a semi-structured document. These scorers can be lexical-based or dense-based, and each field can be scored by both types. We introduce an interpretable and controllable query-conditioned predictor of weights used to adaptively sum over these scores. On three large-scale datasets, we find that MFAR can achieve significant performance gains over existing methods due to multi-field advantages and the inclusion of a hybrid combination of scorers, leading to state-of-the-art performance. Through our analysis, we find that the best models benefit from both access to both scorers and the ability to weight each field conditioned on the query, further verifying our method.

Our primary goal is to study the challenging and emerging problem of retrieval for multi-field semi-structured data and to introduce a flexible framework to approach it. Having laid the groundwork, future work can include more specialized individual scorers, scale up to more scorers in other modalities like vision or audio, and add other algorithmic improvements to the weighted integration of scores across scorers. Then, MFAR would be a step towards retrieval of *any* type of content, which can further aid applications for general search or agent-based RAG.

## 8 ACKNOWLEDGEMENTS

The authors thank the reviewers for the engaging exchange and their helpful feedback. The authors also thank Nick Craswell, Jason Eisner, Jenny Liang, Shirley Wu, Jay DeYoung, Somin Wadhwa, and Byron Wallace, for the insightful discussions to improve the paper. ML conducted this work as a research intern in the MSAI group at Microsoft. ML is also supported by an NSF Graduate Research Fellowship.

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

# A  DATASET

Table 5: The corpus size, number of fields, and queries (by split) for each of the STaRK datasets. For field information, refer to Table 6 in the Appendix.

| Dataset | Domain | Num. Documents | Num. Fields | Train | Dev. | Test. |
|---|---|---|---|---|---|---|
| Amazon | products, product reviews | 950K | 8 | 6K | 1.5K | 1.5K |
| MAG | science papers, authors | 700K | 5 | 8K | 2.6K | 2.6K |
| Prime | biomedical entities | 130K | 22 | 6.1K | 2.2K | 2.8K |

## A.1  PREPROCESSING

Technically, STaRK is a dataset of queries over knowledge graphs. The full dataset details are in Table 5. The baselines (Wu et al., 2024b) create a linearized document for each node, which omits some edge and multi-hop information that is available in the knowledge graph. AvaTaR (Wu et al., 2024a) operates directly on the knowledge graph. As we want to operate over semi-structured documents, we need a preprocessing step either on the linearized documents or by processing the graph.

Because parsing documents is error-prone, we decide to reproduce the document creation process from (Wu et al., 2024b). We start with all of the original dataset from STaRK, which come in the form of queries and the associated answer ids in the knowledge graph. Each query requires a combination of entity information and relation information from their dataset to answer. However, each dataset handles the entity types differently. The answer to every query for Amazon is the `product` entity. For MAG, the answer is the `paper` entity. However, Prime has a list of ten possible entities that can answer the query, so we include all ten as documents.

We create our set of documents based on the directional paths taken in their knowledge graph; if there are more than single hop relations, then we take at most two hops for additional entities and relations. For Amazon, since the queries are at most one hop, we do not include additional node information. MAG and Prime, however, can include more queries with more than two hops, so we include information about additional relations and nodes for each document in our dataset.

## A.2  FIELDS

We include the list of fields that we used in this work in Table 6. Not every single field available in the STaRK knowledge graph (Wu et al., 2024b) is used because some are not used in the baseline and so we try to match the baselines as closely as possible. We make some cosmetic changes for space and clarity in the examples in the main body of this paper, including uppercasing field names and replacing underscore with spaces. We also shorten "author__affiliated_with__institution", "paper__cites__paper" to "Papers Cited", "paper__has_topic__field_of_study" to "Area of Study" and expand "type" to "Entity Type."

Table 6 also lists some information about the length distribution of each field, as measured by the Contriever tokenizer. This is useful to know how much information might be lost to the limited window size of Contriever. Furthermore, we list the maximum sequence length used by the dense scorer of MFAR both during training and at test-time. The trade off for sequence length is batch size with respect to GPU memory usage. Our lexical baseline (BM25) does not perform any truncation.

# B  IMPLEMENTATION DETAILS

During training, we sample $k = 1$ negative example per query. Along with in-batch negatives, this results in $2b - 1$ negative samples for a batch size of $b$. This negative document is sampled using Pyserini Lucene[4]: 100 nearest documents are retrieved, of which the postive documents are removed. The top negative document is then sampled among that remaining set. We apply early

---

[4]https://github.com/castorini/pyserini

Table 6: The datasets and list of fields, $\mathcal{F}$, used in this work, along with basic length statistics of the content of those fields. The length of the $k$th-%ile longest value is listed. For example, $k = 50$ would be the median length. The MSL is the maximum sequence length threshold we chose for that field in MFAR based on either the maximum window size of the encoder (512) or based on covering most (> 99%) documents within the corpus..

| Dataset | Field | Length in Contriever tokens ($k^{\text{th}}$%-ile) | | | | | |
| | | 90 | 95 | 99 | 99.9 | Max | MSL |
|---|---|---|---|---|---|---|---|
| Amazon | also_buy | 64 | 217 | 908 | 3864 | 50176 | 512 |
| | also_view | 86 | 189 | 557 | 1808 | 21888 | 512 |
| | brand | 7 | 8 | 10 | 12 | 35 | 16 |
| | description | 207 | 289 | 446 | 1020 | 5038 | 512 |
| | feature | 130 | 171 | 305 | 566 | 1587 | 512 |
| | qa | 4 | 4 | 5 | 698 | 1873 | 512 |
| | review | 1123 | 2593 | 12066 | 58946 | 630546 | 512 |
| | title | 28 | 34 | 48 | 75 | 918 | 128 |
| MAG | abstract | 354 | 410 | 546 | 775 | 2329 | 512 |
| | author___affiliated_with___institution | 90 | 121 | 341 | 18908 | 46791 | 512 |
| | paper___cites___paper | 581 | 863 | 1785 | 4412 | 79414 | 512 |
| | paper___has_topic___field_of_study | 49 | 52 | 57 | 63 | 90 | 64 |
| | title | 31 | 34 | 44 | 62 | 9934 | 64 |
| Prime | associated with | 10 | 35 | 173 | 706 | 4985 | 256 |
| | carrier | 3 | 4 | 4 | 13 | 2140 | 8 |
| | contraindication | 4 | 4 | 66 | 586 | 3481 | 128 |
| | details | 329 | 823 | 2446 | 5005 | 12319 | 512 |
| | enzyme | 4 | 4 | 12 | 63 | 5318 | 64 |
| | expression absent | 4 | 8 | 29 | 77 | 12196 | 64 |
| | expression present | 204 | 510 | 670 | 18306 | 81931 | 512 |
| | indication | 4 | 4 | 25 | 146 | 1202 | 32 |
| | interacts with | 93 | 169 | 446 | 1324 | 55110 | 512 |
| | linked to | 3 | 4 | 4 | 57 | 544 | 8 |
| | name | 17 | 21 | 38 | 74 | 133 | 64 |
| | off-label use | 3 | 4 | 4 | 56 | 727 | 8 |
| | parent-child | 49 | 70 | 168 | 714 | 18585 | 256 |
| | phenotype absent | 3 | 4 | 4 | 33 | 1057 | 8 |
| | phenotype present | 20 | 82 | 372 | 1931 | 28920 | 512 |
| | ppi | 36 | 125 | 438 | 1563 | 22432 | 512 |
| | side effect | 4 | 4 | 93 | 968 | 5279 | 128 |
| | source | 5 | 6 | 6 | 7 | 8 | 8 |
| | synergistic interaction | 4 | 4 | 4800 | 9495 | 13570 | 512 |
| | target | 4 | 9 | 33 | 312 | 5852 | 64 |
| | transporter | 3 | 4 | 4 | 41 | 2721 | 8 |
| | type | 7 | 8 | 8 | 9 | 9 | 8 |

stopping on validation loss with a patience of 5. We set $\tau = 0.05$ and train with DDP on 8x NVIDIA A100s. Contriever is a 110M parameter model, and the additional parameters added through $G$ is negligible ($768|\mathcal{F}|$), scaling linearly in the number of fields.

We use separate learning rates (LRs) for finetuning the encoder and for the other parameters. Specifically, we searched over learning rates [5e-6, **1e-5, 5e-5**, 1e-4] for the encoder and [1e-3, 5e-3, **1e-2**, **5e-2**, 1e-1] for the parameters in $G(q, f, m)$ which consist of $\mathbf{a}_f^m$ and $\gamma_f^m$ and $\beta_f^m$ from batch normalization. The main grid search was conducted over the bolded values, although we found 5e-3 to be effective for $G(q, f, m)$ for Amazon. We otherwise follow the default settings for both the optimizer (AdamW, dropout, etc.) and batch normalization (PyTorch 2.4.0). As mentioned, whether to apply batch normalization at all was also a hyperparameter searched over: we found it useful in the hybrid setting.

Our implementation uses Pytorch Lightning[5] and `sentence-transformers` 2.2.2 (Reimers & Gurevych, 2019). We use a fast, python-based implementation of BM25 as our lexical scorer (Lù, 2024).[6] The best hyperparameters for each of our models in this work are listed in Table 7. In the case where there is only a single field (last two sections), the adaptive query conditioning is not needed.

At inference, we retrieve the top-100 results per field to form a candidate set, and we compute the full scores over this candidate set to obtain our final ranking.

Table 7: The hyperparameters used for each of the runs in this work.

| Model | Dataset | Mainly referenced in | Encoder LR | $G()$ LR | Batch norm? |
|---|---|---|---|---|---|
| $\text{MFAR}_{\text{All}}$ | Amazon | Table 1, most tables/figures | 1e-5 | 5e-3 | no |
| | MAG | | 5e-5 | 1e-2 | yes |
| | Prime | | 5e-5 | 1e-2 | yes |
| $\text{MFAR}_{\text{Dense}}$ | Amazon | Table 1, Figure 3 | 1e-5 | 5e-3 | no |
| | MAG | | 5e-5 | 5e-2 | no |
| | Prime | | 1e-5 | 1e-2 | no |
| $\text{MFAR}_{\text{Lexical}}$ | Amazon | Table 1, Figure 3 | 1e-5 | 5e-3 | yes |
| | MAG | | 1e-5 | 1e-2 | yes |
| | Prime | | 5e-5 | 1e-1 | yes |
| $\text{MFAR}_2$ | Amazon | Table 1, Figure 3 | 1e-5 | 1e-2 | no |
| | MAG | | 5e-5 | 5e-3 | yes |
| | Prime | | 5e-5 | 5e-3 | yes |
| $\text{MFAR}_{\text{Dense\&1}}$ | Amazon | Table 10 (appendix only) | 1e-5 | 1e-2 | no |
| | MAG | | 5e-5 | 5e-3 | yes |
| | Prime | | 1e-5 | 5e-3 | yes |
| Contriever-FT | Amazon | Table 1 | 5e-5 | n/a | n/a |
| | MAG | | 1e-5 | n/a | n/a |
| | Prime | | 5e-5 | n/a | n/a |

## C    DETAILED RESULTS ON STARK

We present comprehensive results on the test split of Amazon, MAG, and Prime.

### C.1    FULL TEST RESULTS AND COMPARISON

Here, we report the same results as in the main section, but we also include H@5, to be exhaustive with STaRK, in addition to our existing metrics. In Table 8, we show the test results with the included additional metric. In Table 9, we also include the average as a separate table. Here, we find that MFAR still does on average better than the other baselines on the semi-structured datasets, even against the strong lexical BM25 baseline.

### C.2    MERGING FULL WITH PER-FIELD REPRESENTATIONS

We evaluate additional models that make combine both multi-field ($|\mathcal{F}|$ scorers) with a single-field (one concatenated document) document representation for either or both lexical and dense retrievers. For example, BM25 could be interpreted as a lexical retriever over a (single-field) full-document text which concatenates all the fields. We extend our earlier experiments with 4 additional experiments: $\text{MFAR}_{\text{Lexical+1}}$ is $\text{MFAR}_{\text{Lexical}}$ with an additional scorer (field) that scores the full document by using the single-document BM25 score; $\text{MFAR}_{\text{Dense+1}}$ is $\text{MFAR}_{\text{Dense}}$ with an additional scorer that embeds and scores the full document (using finetuned Contriever); $\text{MFAR}_{\text{All+2}}$ is a hybrid model, like $\text{MFAR}_{\text{All}}$, which combines $\text{MFAR}_{\text{Lexical+1}}$ and $\text{MFAR}_{\text{Dense+1}}$, i.e. this contains the most scorers.

---

[5]https://lightning.ai/
[6]https://github.com/xhluca/bm25s

Table 8: Similar to Table 1, we instead include H@5 and show the average as a separate table, over the test split. We include the same baselines and generally find that H@5 also follows the same trends. The average over all datasets can be seen in Table 9.

| Model | Amazon | | | | MAG | | | | Prime | | | |
|---|---|---|---|---|---|---|---|---|---|---|---|---|
| | H@1 | H@5 | R@20 | MRR | H@1 | H@5 | R@20 | MRR | H@1 | H@5 | R@20 | MRR |
| ada-002 | 0.392 | 0.627 | 0.533 | 0.542 | 0.291 | 0.496 | 0.484 | 0.386 | 0.126 | 0.315 | 0.360 | 0.214 |
| multi-ada-002 | 0.401 | 0.650 | 0.551 | 0.516 | 0.259 | 0.504 | 0.508 | 0.369 | 0.151 | 0.336 | 0.381 | 0.235 |
| Claude3 | 0.455 | 0.711 | 0.538 | 0.559 | 0.365 | 0.532 | 0.484 | 0.442 | 0.178 | 0.369 | 0.356 | 0.263 |
| GPT4 | 0.448 | 0.712 | 0.554 | 0.557 | 0.409 | 0.582 | 0.486 | 0.490 | 0.183 | 0.373 | 0.341 | 0.266 |
| BM25 | 0.483 | 0.721 | 0.584 | 0.589 | 0.471 | 0.693 | 0.689 | 0.572 | 0.167 | 0.355 | 0.410 | 0.255 |
| Contriever-FT | 0.383 | 0.639 | 0.530 | 0.497 | 0.371 | 0.594 | 0.578 | 0.475 | 0.325 | 0.548 | 0.600 | 0.427 |
| AvaTaR (agent) | 0.499 | 0.692 | 0.606 | 0.587 | 0.444 | 0.567 | 0.506 | 0.512 | 0.184 | 0.367 | 0.393 | 0.267 |
| $\text{MFAR}_{\text{Lexical}}$ | 0.332 | 0.569 | 0.491 | 0.443 | 0.429 | 0.634 | 0.657 | 0.522 | 0.257 | 0.455 | 0.500 | 0.347 |
| $\text{MFAR}_{\text{Dense}}$ | 0.390 | 0.659 | 0.555 | 0.512 | 0.467 | 0.678 | 0.669 | 0.564 | 0.375 | 0.620 | 0.698 | 0.485 |
| $\text{MFAR}_2$ | **0.574** | **0.814** | **0.663** | **0.681** | 0.503 | 0.717 | 0.721 | 0.603 | 0.227 | 0.439 | 0.495 | 0.327 |
| $\text{MFAR}_{\text{All}}$ | 0.412 | 0.700 | 0.585 | 0.542 | 0.490 | 0.696 | 0.717 | 0.582 | **0.409** | 0.628 | 0.683 | 0.512 |
| $\text{MFAR}_{\text{All+2}}$ | 0.530 | 0.785 | **0.663** | 0.643 | **0.559** | **0.742** | **0.741** | **0.643** | 0.400 | **0.659** | **0.726** | **0.520** |

Table 9: The averages for Table 8.

| Model | Averages | | | |
|---|---|---|---|---|
| | H@1 | H@5 | R@20 | MRR |
| ada-002 | 0.270 | 0.479 | 0.459 | 0.381 |
| multi-ada-002 | 0.270 | 0.497 | 0.480 | 0.373 |
| Claude3* | 0.333 | 0.537 | 0.459 | 0.421 |
| GPT4* | 0.347 | 0.556 | 0.460 | 0.465 |
| BM25 | 0.374 | 0.590 | 0.561 | 0.462 |
| Contriever-FT | 0.360 | 0.594 | 0.569 | 0.467 |
| AvaTaR (agent) | 0.376 | 0.542 | 0.502 | 0.455 |
| $\text{MFAR}_{\text{Lexical}}$ | 0.339 | 0.553 | 0.549 | 0.437 |
| $\text{MFAR}_{\text{Dense}}$ | 0.411 | 0.652 | 0.641 | 0.520 |
| $\text{MFAR}_2$ | 0.435 | 0.656 | 0.626 | 0.537 |
| $\text{MFAR}_{\text{All}}$ | 0.437 | 0.675 | 0.662 | 0.545 |
| $\text{MFAR}_{\text{All+2}}$ | **0.496** | **0.729** | **0.710** | **0.602** |

Finally, $\text{MFAR}_{\text{Dense\&1}}$ is a hybrid model that combines $\text{MFAR}_{\text{Dense}}$ with an additional BM25 score over the full single-document representation. We separately experiment with this combination due to the promising results from the BM25 baseline (compared to $\text{MFAR}_{\text{Lexical}}$) over certain datasets, the relative strength of $\text{MFAR}_{\text{Dense}}$ compared to $\text{MFAR}_{\text{All}}$, and our qualitative analysis.

Each of these models re-used the hyperparameters from their corresponding base version. The one exception is $\text{MFAR}_{\text{Dense\&1}}$, for which we performed an additional grid search like the earlier models as there is no clear corresponding base model.

Comparing these scores against the base model scores (from Table 1), we find that there is considerable benefit to including a single-document representation in addition to the multi-field one.

### C.3 VARIANCE ACROSS RANDOM SEEDS

For each trained model in Table 1, we select 3 additional random seeds and retrain those models to establish the variation of the scores across runs. Note that none of these seeds are the same as the one used in our main experiments. Notably, $\text{MFAR}_{\text{Dense\&1}}$ outperforms $\text{MFAR}_{\text{All}}$ despite having fewer scorers ($|\mathcal{F}| + 1$ vs. $2|\mathcal{F}|$)

## D  BM25F

BM25F is included as a point of comparison in our work, but there exists no public implementation in Python, so we manually implement BM25F using an existing codebase [7]. We do not exhaustively

---
[7] https://github.com/jxmorris12/bm25_pt

Table 10: Scores of models that consider a combination of single-field and multi-field document representations.

| | Amazon | | | | MAG | | | | Prime | | | |
|---|---|---|---|---|---|---|---|---|---|---|---|---|
| Model | H@1 | H@5 | R@20 | MRR | H@1 | H@5 | R@20 | MRR | H@1 | H@5 | R@20 | MRR |
| $\text{MFAR}_{\text{Lexical}}$ | 0.332 | 0.569 | 0.491 | 0.443 | 0.429 | 0.634 | 0.657 | 0.522 | 0.257 | 0.455 | 0.500 | 0.347 |
| $\text{MFAR}_{\text{Lexical+1}}$ | 0.471 | 0.728 | 0.605 | 0.588 | 0.470 | 0.675 | 0.690 | 0.564 | 0.271 | 0.479 | 0.541 | 0.367 |
| $\text{MFAR}_{\text{Dense}}$ | 0.390 | 0.659 | 0.555 | 0.512 | 0.467 | 0.678 | 0.669 | 0.564 | 0.375 | 0.620 | 0.698 | 0.485 |
| $\text{MFAR}_{\text{Dense+1}}$ | 0.453 | 0.704 | 0.594 | 0.570 | 0.472 | 0.703 | 0.679 | 0.576 | 0.384 | 0.636 | 0.707 | 0.498 |
| $\text{MFAR}_{\text{All}}$ | 0.412 | 0.700 | 0.585 | 0.542 | 0.490 | 0.696 | 0.717 | 0.582 | 0.409 | 0.628 | 0.683 | 0.512 |
| $\text{MFAR}_{\text{Dense\&1}}$ | 0.530 | 0.785 | 0.663 | 0.643 | 0.559 | 0.742 | 0.741 | 0.643 | 0.400 | 0.659 | 0.726 | 0.520 |
| $\text{MFAR}_{\text{All+2}}$ | 0.562 | 0.808 | 0.672 | 0.674 | 0.507 | 0.721 | 0.717 | 0.605 | 0.342 | 0.615 | 0.669 | 0.464 |

Table 11: Standard Deviation of each metric for each dataset and model. These are typically between 0 to 0.015, which gives a sense of how significant differences between models are.

| | Amazon | | | | MAG | | | | Prime | | | |
|---|---|---|---|---|---|---|---|---|---|---|---|---|
| Model | H@1 | H@5 | R@20 | MRR | H@1 | H@5 | R@20 | MRR | H@1 | H@5 | R@20 | MRR |
| $\text{MFAR}_{\text{Lexical}}$ | 0.009 | 0.007 | 0.004 | 0.008 | 0.014 | 0.008 | 0.002 | 0.001 | 0.013 | 0.009 | 0.016 | 0.013 |
| $\text{MFAR}_{\text{Dense}}$ | 0.007 | 0.004 | 0.004 | 0.003 | 0.008 | 0.005 | 0.001 | 0.001 | 0.005 | 0.003 | 0.004 | 0.002 |
| $\text{MFAR}_2$ | 0.004 | 0.003 | 0.004 | 0.003 | 0.010 | 0.007 | 0.007 | 0.001 | 0.010 | 0.014 | 0.012 | 0.002 |
| $\text{MFAR}_{\text{All}}$ | 0.001 | 0.002 | 0.002 | 0.001 | 0.026 | 0.020 | 0.020 | 0.007 | 0.014 | 0.011 | 0.011 | 0.012 |
| $\text{MFAR}_{\text{Dense\&1}}$ | 0.001 | 0.002 | 0.001 | 0.001 | 0.011 | 0.009 | 0.009 | 0.002 | 0.004 | 0.002 | 0.002 | 0.003 |
| $\text{MFAR}_{\text{All+2}}$ | 0.012 | 0.007 | 0.007 | 0.001 | 0.005 | 0.009 | 0.009 | 0.005 | 0.006 | 0.012 | 0.012 | 0.008 |

search across weights to set, as it would require as many as $2|\mathcal{F}|+1$ independent parameter searches (Zaragoza et al., 2004). As a result, an exhaustive BM25F baseline with optimal weights is difficult without large amounts of compute[8], whereas MFAR does not require such a grid search that scales in the number of fields of the dataset.

Below, we present BM25F with uniform weights for each field, setting all weights to 1, compared to regular BM25 and $\text{MFAR}_{\text{All}}$. Though there may be more optimal settings, the weight selection is entirely dataset-dependent.

Table 12: A uniformly-weighted BM25F against BM25 and $\text{MFAR}_{\text{All}}$.

| | Amazon | | | MAG | | | Prime | | |
|---|---|---|---|---|---|---|---|---|---|
| Model | H@1 | R@20 | MRR | H@1 | R@20 | MRR | H@1 | R@20 | MRR |
| BM25 | 0.483 | 0.584 | 0.589 | 0.471 | 0.689 | 0.472 | 0.167 | 0.410 | 0.255 |
| BM25F | 0.183 | 0.332 | 0.264 | 0.451 | 0.671 | 0.551 | 0.142 | 0.244 | 0.214 |
| $\text{MFAR}_{\text{All}}$ | 0.412 | 0.585 | 0.542 | 0.490 | 0.717 | 0.582 | 0.409 | 0.683 | 0.512 |

We find that using uniform weights, BM25F performs even worse than BM25. This highlights the importance of choosing appropriate weights, which is nontrivial. In contrast, the relative importance (or weights) assigned to each field is learned in MFAR.

---

[8]There exist methods such as RankLib (John Foley, 2019), a learning-to-rank algorithm, and coordinate ascent, that can also be used for searching weights, but we find that this still requires a large amount of compute to fully realize. For each query and document pair, one must generate a set of features to be used for RankLib. These features scale with the number of documents. Therefore, if the combination of queries and documents is large, then generating all possible features may become intractable. Additionally, if one chooses to add more samples, it is nontrivial to then use RankLib again (one would have to search again from scratch).

## E    FULL RESULTS FOR FIELD MASKING

We include full scores for masking each field and scorer for Amazon in Table 13, MAG in Table 14, and Prime in Table 15. The first row "—" is MFAR$_{\text{All}}$ without any masking and repeated three times as a reference. The final row "all" is the result of masking out all the lexical scores (or all the dense scores). It does not make sense to mask out all scores, as that would result in no scorer.

Based on our findings in Table 14, all fields in MAG are generally useful, as all instances of zeroing out the respect fields results in a performance drop. Despite this finding with MAG, not all fields are as obviously important in other datasets. For Table 15, Prime has a notable number of fields that do not contribute to the final ranking when both scorers are masked out. And for Amazon, in Table 13, we surprisingly find that fields like "description" and "brand" have little effect. This is a reflection on both the dataset (and any redundancies contained within) and on the distribution of queries and what they ask about.

Table 13: Test scores on Amazon after masking out each field and scorer of the MFAR$_{\text{All}}$ at test-time.

| Amazon | $w_f^{\text{lexical}} = 0$ | | | | $w_f^{\text{dense}} = 0$ | | | | Both | | | |
|---|---|---|---|---|---|---|---|---|---|---|---|---|
| Masked field | H@1 | H@5 | R@20 | MRR | H@1 | H@5 | R@20 | MRR | H@1 | H@5 | R@20 | MRR |
| — | 0.412 | 0.700 | 0.586 | 0.542 | 0.412 | 0.700 | 0.586 | 0.542 | 0.412 | 0.700 | 0.586 | 0.542 |
| also_buy | 0.407 | 0.690 | 0.578 | 0.534 | 0.410 | 0.696 | 0.586 | 0.540 | 0.403 | 0.678 | 0.578 | 0.530 |
| also_view | 0.420 | 0.695 | 0.576 | 0.542 | 0.414 | 0.696 | 0.581 | 0.542 | 0.395 | 0.677 | 0.565 | 0.522 |
| brand | 0.410 | 0.699 | 0.585 | 0.540 | 0.397 | 0.692 | 0.575 | 0.528 | 0.400 | 0.686 | 0.570 | 0.526 |
| description | 0.417 | 0.699 | 0.587 | 0.542 | 0.410 | 0.692 | 0.580 | 0.540 | 0.413 | 0.680 | 0.576 | 0.535 |
| feature | 0.412 | 0.700 | 0.581 | 0.537 | 0.398 | 0.680 | 0.570 | 0.524 | 0.410 | 0.680 | 0.562 | 0.531 |
| qa | 0.412 | 0.700 | 0.586 | 0.542 | 0.412 | 0.700 | 0.586 | 0.542 | 0.381 | 0.636 | 0.545 | 0.499 |
| review | 0.410 | 0.696 | 0.583 | 0.541 | 0.398 | 0.680 | 0.575 | 0.526 | 0.384 | 0.666 | 0.548 | 0.510 |
| title | 0.414 | 0.685 | 0.583 | 0.535 | 0.390 | 0.650 | 0.555 | 0.508 | 0.389 | 0.672 | 0.562 | 0.516 |
| all | 0.389 | 0.660 | 0.553 | 0.512 | 0.271 | 0.518 | 0.452 | 0.386 | — | — | — | — |

Table 14: Test scores on MAG after masking out each field and scorer of the MFAR$_{\text{All}}$ at test-time. Due to space, we truncate some field names, refer to Table 6 for the full names.

| MAG | $w_f^{\text{lexical}} = 0$ | | | | $w_f^{\text{dense}} = 0$ | | | | Both | | | |
|---|---|---|---|---|---|---|---|---|---|---|---|---|
| Masked field | H@1 | H@5 | R@20 | MRR | H@1 | H@5 | R@20 | MRR | H@1 | H@5 | R@20 | MRR |
| — | 0.490 | 0.696 | 0.717 | 0.582 | 0.490 | 0.696 | 0.717 | 0.582 | 0.490 | 0.696 | 0.717 | 0.582 |
| abstract | 0.469 | 0.681 | 0.707 | 0.565 | 0.393 | 0.616 | 0.651 | 0.494 | 0.430 | 0.636 | 0.659 | 0.526 |
| author affil... | 0.338 | 0.555 | 0.600 | 0.439 | 0.490 | 0.696 | 0.717 | 0.582 | 0.389 | 0.595 | 0.631 | 0.485 |
| paper cites... | 0.458 | 0.660 | 0.655 | 0.551 | 0.484 | 0.685 | 0.708 | 0.576 | 0.424 | 0.650 | 0.668 | 0.526 |
| paper topic... | 0.459 | 0.671 | 0.695 | 0.554 | 0.491 | 0.695 | 0.717 | 0.582 | 0.398 | 0.617 | 0.650 | 0.499 |
| title | 0.479 | 0.686 | 0.714 | 0.573 | 0.473 | 0.676 | 0.703 | 0.565 | 0.414 | 0.633 | 0.654 | 0.513 |
| all | 0.257 | 0.462 | 0.481 | 0.355 | 0.352 | 0.561 | 0.602 | 0.446 | — | — | — | — |

## F    TABLE RETRIEVAL

In Table 16, we demonstrate MFAR on table retrieval, specifically on NQ-Tables (Herzig et al., 2021) which consists of 170K tables along with their titles. We note the dataset has generally short inputs (with limited decomposition of fields into title, column headers, and table content), where fine-tuned full-context models may excel over MFAR. The dataset is sourced from Wikipedia, which contains knowledge seen in pretraining data. Finally, we did not sweep hyperparameters - instead re-using those from earlier. We compare to DPR-table, a model of similar size finetuned over tables (Wang et al., 2022). DPR-table outperforms MFAR by at large margin at Hit@1. However, we find that MFAR has improved recall over the table retrieval model when considering top-10 or top-20 results. This shows that even in mismatched tasks (where there are few fields and a competitive baseline designed for those fields), MFAR can show promise.

Table 15: Test scores on Prime after masking out each field and scorer of the MFAR$_{\text{All}}$ at test-time. Due to space, we shorten some field names, so refer to Table 6 for the full names.

| Prime | $w_f^{\text{lexical}} = 0$ | | | | $w_f^{\text{dense}} = 0$ | | | | Both | | | |
|---|---|---|---|---|---|---|---|---|---|---|---|---|
| Masked field | H@1 | H@5 | R@20 | MRR | H@1 | H@5 | R@20 | MRR | H@1 | H@5 | R@20 | MRR |
| — | 0.409 | 0.627 | 0.683 | 0.512 | 0.409 | 0.627 | 0.683 | 0.512 | 0.409 | 0.627 | 0.683 | 0.512 |
| associated with | 0.392 | 0.610 | 0.670 | 0.495 | 0.407 | 0.624 | 0.680 | 0.510 | 0.399 | 0.618 | 0.672 | 0.502 |
| carrier | 0.409 | 0.627 | 0.683 | 0.512 | 0.409 | 0.627 | 0.683 | 0.512 | 0.403 | 0.621 | 0.678 | 0.506 |
| contraindication | 0.409 | 0.627 | 0.683 | 0.512 | 0.409 | 0.627 | 0.683 | 0.512 | 0.380 | 0.587 | 0.652 | 0.479 |
| details | 0.386 | 0.606 | 0.670 | 0.488 | 0.363 | 0.569 | 0.619 | 0.458 | 0.388 | 0.601 | 0.661 | 0.489 |
| enzyme | 0.409 | 0.627 | 0.683 | 0.512 | 0.409 | 0.627 | 0.683 | 0.512 | 0.405 | 0.623 | 0.677 | 0.508 |
| expression abs. | 0.408 | 0.627 | 0.683 | 0.511 | 0.392 | 0.607 | 0.664 | 0.494 | 0.403 | 0.622 | 0.678 | 0.506 |
| expression pres. | 0.409 | 0.627 | 0.683 | 0.512 | 0.409 | 0.627 | 0.683 | 0.512 | 0.400 | 0.617 | 0.675 | 0.502 |
| indication | 0.407 | 0.627 | 0.682 | 0.511 | 0.398 | 0.613 | 0.663 | 0.498 | 0.392 | 0.611 | 0.665 | 0.495 |
| interacts with | 0.403 | 0.624 | 0.681 | 0.507 | 0.406 | 0.626 | 0.682 | 0.510 | 0.403 | 0.622 | 0.674 | 0.506 |
| linked to | 0.409 | 0.627 | 0.683 | 0.512 | 0.409 | 0.627 | 0.683 | 0.512 | 0.383 | 0.601 | 0.661 | 0.486 |
| name | 0.410 | 0.628 | 0.684 | 0.513 | 0.407 | 0.627 | 0.681 | 0.510 | 0.407 | 0.622 | 0.674 | 0.507 |
| off-label use | 0.409 | 0.627 | 0.683 | 0.512 | 0.409 | 0.627 | 0.683 | 0.512 | 0.379 | 0.602 | 0.662 | 0.482 |
| parent-child | 0.385 | 0.619 | 0.680 | 0.494 | 0.391 | 0.613 | 0.663 | 0.495 | 0.386 | 0.601 | 0.663 | 0.487 |
| phenotype abs. | 0.408 | 0.625 | 0.681 | 0.511 | 0.409 | 0.627 | 0.683 | 0.512 | 0.376 | 0.591 | 0.653 | 0.477 |
| phenotype pres. | 0.405 | 0.619 | 0.675 | 0.506 | 0.409 | 0.627 | 0.683 | 0.512 | 0.393 | 0.609 | 0.669 | 0.495 |
| ppi | 0.403 | 0.622 | 0.678 | 0.506 | 0.409 | 0.627 | 0.683 | 0.512 | 0.399 | 0.617 | 0.671 | 0.502 |
| side effect | 0.409 | 0.627 | 0.683 | 0.512 | 0.409 | 0.627 | 0.683 | 0.512 | 0.405 | 0.624 | 0.680 | 0.508 |
| source | 0.409 | 0.627 | 0.683 | 0.512 | 0.409 | 0.627 | 0.683 | 0.512 | 0.397 | 0.614 | 0.671 | 0.499 |
| synergistic int. | 0.408 | 0.627 | 0.682 | 0.511 | 0.409 | 0.627 | 0.683 | 0.512 | 0.381 | 0.597 | 0.659 | 0.483 |
| target | 0.407 | 0.627 | 0.683 | 0.511 | 0.394 | 0.613 | 0.662 | 0.497 | 0.397 | 0.617 | 0.671 | 0.501 |
| transporter | 0.409 | 0.627 | 0.683 | 0.512 | 0.409 | 0.627 | 0.683 | 0.512 | 0.406 | 0.624 | 0.679 | 0.509 |
| type | 0.409 | 0.627 | 0.683 | 0.512 | 0.403 | 0.625 | 0.681 | 0.507 | 0.396 | 0.615 | 0.669 | 0.498 |
| all | 0.342 | 0.554 | 0.624 | 0.442 | 0.267 | 0.450 | 0.500 | 0.352 | — | — | — | — |

Table 16: Table retrieval results on NQ-Tables (Herzig et al., 2021). We report recall, since there is only one gold document per query.

| Model | R@1 | R@5 | R@10 | R@20 |
|---|---|---|---|---|
| MFAR$_2$ | 0.497 | 0.812 | 0.878 | 0.930 |
| MFAR$_{\text{All}}$ | 0.498 | 0.829 | 0.900 | 0.949 |
| DPR-table 110M | 0.679 | 0.849 | 0.889 | 0.906 |

