# OpenReview forum: "Multi-Field Adaptive Retrieval"
_ICLR.cc/2025/Conference — ICLR 2025 Spotlight_

### Official Review · Reviewer_Hpub · 2024-11-03

**Soundness:** 2
**Presentation:** 3
**Contribution:** 2
**Rating:** 6
**Confidence:** 5

**Summary:**

This paper introduces a method called Multi-Field Adaptive Retrieval (MFAR), aimed at enhancing retrieval of structured or multi-field documents. The approach decomposes each document into multiple fields, each independently indexed and retrievable through different models. Each field is associated with a learnable embedding, and an adaptive function is trained to predict the importance of each field based on the query and scoring model. The authors validate MFAR's effectiveness through experiments across multiple datasets, comparing it against prior methods.

**Strengths:**

- S1: This study is the first to demonstrate that multi-field ranking outperforms single-field ranking within the context of dense retrieval.
- S2: The experiments highlight that hybrid models enhance both single- and multi-field rankings, with certain multi-field scenarios benefitting more from hybrid approaches.
- S3: A thorough ablation study demonstrates the significance of the query-conditioning mechanism in optimizing field weights. Additionally, the qualitative analysis shows the variability in optimal scorers across datasets, showing that there is no universally best approach.
- S4: A detailed case study further illustrates the method’s effectiveness in practical applications.

**Weaknesses:**

- W1: Although the paper focuses on multi-field ranking, it does not include classic methods such as BM25F [1], its later extensions [2,3], or the mixture of language models [5], which are commonly applied in table retrieval [6], as part of the baselines. Incorporating at least BM25F or the mixture of language models would add valuable context and enable a more thorough comparison.
- W2: Query-dependent field weighting has been previously explored, such as in table retrieval methods that incorporate both query-dependent and query-independent features [4]. Testing on table retrieval datasets could offer additional insights, as tables represent another structured, multi-field document type.
- W3: The proposed method adaptively determines the importance of each field given a query and scorer; however, it does not select among scorers, instead requiring calculation of all scoring potentials, thereby increasing computational load.


[1] Simple BM25 extension to multiple weighted fields, 2004

[2] Field-Weighted XML Retrieval Based on BM25, 2006

[3] Extending BM25 with Multiple Query Operators, 2012

[4] Web Table Retrieval using Multimodal Deep Learning, 2020

[5] Combining Document Representations for Known-Item Search, SIGIR 2003

[6] Ad Hoc Table Retrieval using Semantic Similarity, WWW 2018

**Questions:**

n/a

---

> ### Author Response · Authors · 2024-11-23
> **Response to reviewer, pt1**
>
> We thank the reviewer for their comprehensive feedback. We detail our responses below:
>
> > W1: Although the paper focuses on multi-field ranking, classic methods like BM25F [1] and its subsequent extensions [2,3] are not included in the baselines. Including at least BM25F would provide valuable context and facilitate a more comprehensive comparison.
>
> Thanks for the suggestion, and we will include a BM25F baseline. However, we’d like to emphasize that in BM25F, we would need to pre-select (or search) the weights for BM25F and these weights would be constant for each query. And, we’ve found that without query conditioning, scores are considerably worse (Table 3). Nonetheless, this week we implemented BM25F [1] on the documents, and as a starting point, picked weights of 1 for each field on Amazon, MAG, and Prime, and obtained the following scores (we repeat the BM25 and mFAR_all scores below too), just to get a sense of how it compares. Since we have no prior understanding for what weights may be best, we adopted a uniform prior across fields by setting each weight to 1. This is another limitation of using BM25F: prior works had very few fields [1] and could hand select weights (mainly HTML webpages, so 3 or fewer fields); in our use case, the weight selection must be made in advanced, and do not know the importance of each field ahead of time (the best we can do would be intuition based on a cursory look).
>
> |     | Amazon - H@1 | R@20 | MRR | MAG - H @1 | R@20 | MRR | Prime - H@1 | R@20 | MRR |
> | -------- | ------- | ------- | -------- | ------- | ------- | -------- | ------- | ------- | ------- |
> |BM25 | 0.483 | 0.584 | 0.589 | 0.471 | 0.689 | 0.572 | 0.167 | 0.410 | 0.255 |
> |BM25F | 0.183 | 0.332 | 0.264 | 0.451 | 0.671 | 0.551 | 0.142 | 0.244 | 0.214 |
> |mFAR$_{all}$ | 0.412 | 0.585 | 0.542 | 0.490 | 0.717 | 0.582 | 0.409 | 0.683 | 0.512 |
>
> BM25F fails to perform well, and possibly in the case of MAG, uniform weights happen to do quite well. Furthermore, BM25F does not see gains over BM25, making it a weaker baseline.
>
> Going further, one may try to learn the BM25F weights end-to-end in combination with the query adaptation in mFAR, but that would require a GPU-friendly version of BM25F, which we are not aware of, and we did not have time to implement it this week. We think it could be interesting to try, however.
>
> [1] Simple BM25 extension to multiple weighted fields, https://dl.acm.org/doi/10.1145/1031171.1031181, 2004

---

> > ### Author Response · Authors · 2024-11-23
> > **Response to reviewer, pt2**
> >
> > > W2: Query-dependent field weighting has been previously explored, such as in table retrieval methods that incorporate both query-dependent and query-independent features [4]. Testing on table retrieval datasets could offer additional insights, as tables represent another structured, multi-field document type.
> >
> > Thanks for raising this. First, we want to emphasize that our approach does not discriminate between different types of fields, while in [4], they use a different architecture for each field and [5] uses Wikipedia-specific information. These must be made in advance based on the characteristics of each field.
> >
> > We agree table retrieval is related to (semi-)structured retrieval, although some researchers disagree (see last paragraph of our response to this question). However, the datasets used by past work [4] are not suitable for us: one of them is no longer available [5], while the other consists of only 60 queries [6], which we do not expect will be enough for meaningful experiments. Another tabular datasets [7] contain only tables without additional fields or titles, so we do not think that is suitable either.
> >
> > We found another recent table retrieval dataset (NQ Tables [8]) which consists of 170K tables along with their titles. We stress that we do not expect this dataset to demonstrate the strengths of our method because: 1) we can only decompose it into 3 fields: title, column headers, table content and most tables are small enough to fully fit in the encoder context; 2) these tables are from Wikipedia and seen in pretraining for DPR/DPR-table [10] and only partially for Contriever; 3) we did not sweep hyperparameters; and 4) [9] argues that this dataset (and table retrieval in general) is not a structure problem.
> >
> > As requested, here are some results for mFAR_all and mFAR_2 alongside the best numbers from [9], a fine-tuned passage retriever model. There is only one gold (relevant) document per query, so we only report recall.
> >
> > | | R@1 | R@5 | R@10 | R@15 | R@20 |
> > | -------- | ------- | ------- | -------- | ------- | ------- |
> > |mFAR_all  |  0.497 | 0.812 | 0.878 | 0.915 | 0.930 |
> > |mFAR_2    |   0.498 | 0.829 | 0.900 | 0.933 | 0.949 |
> > | DPR-table 110M [9] | 0.679 | 0.849 | 0.889 | –  | 0.906 |
> >
> > Again, these were put together this week, and when we add this discussion into our paper as another example dataset, we will have time to double-check. We find that our method has generally higher recall but worse recall/hit in top-1.
> >
> > Finally, [9] main conclusion is that “In summary, our analysis reveals that understanding table structure is not necessary in the majority of cases.” If this is the case, then this specific table retrieval dataset is more like web text retrieval and this dataset, like MSMARCO, is not suitable for evaluating the strengths of our model. This brings us back to the start, where we could not find a suitable table retrieval dataset.
> >
> > [2] Semantic Table Retrieval using Keyword and Table Queries, https://arxiv.org/abs/2105.06365, 2021
> >
> > [4] Web Table Retrieval using Multimodal Deep Learning, https://dl.acm.org/doi/10.1145/3397271.3401120, 2020
> >
> > [5] https://github.com/haggair/gnqtables (no longer available)
> >
> > [6] TabEL: entity linking in web tables, https://link.springer.com/chapter/10.1007/978-3-319-25007-6_25,  2015
> >
> > [7] Compositional Semantic Parsing on Semi-Structured Tables, https://arxiv.org/abs/1508.00305, 2015
> >
> > [8] Open Domain Question Answering over Tables via Dense Retrieval, https://arxiv.org/abs/2103.12011, 2021
> >
> > [9] Table Retrieval May Not Necessitate Table-specific Model Design, https://arxiv.org/abs/2205.09843, 2022
> >
> > [10] Dense Passage Retrieval for Open-Domain Question Answering, https://arxiv.org/abs/2004.04906, 2020
> >
> > > W3: The proposed method adaptively determines the importance of each field given a query and scorer; however, it does not select among scorers, instead requiring calculation of all scoring potentials, thereby increasing computational load.
> >
> > We think that it is important to allow our framework to learn from the n different scorers as it is not determined beforehand which of the n scorers for any single field will be most useful with respect to our query set. For future work, we agree that our method would benefit from pruning approaches where we remove fields that consistently have low weights.
> >
> > Empirically, our method did not significantly slow down training. Computing embeddings across fields can be batched/parallelized, and because the text within fields are shorter than the full doc, the embedding might even be faster. For the mFAR_dense vs. Contriever FT, Amazon is about 1.6x slower, Prime is 3.3x slower, and MAG is 2.2x slower. Our sparse scoring code (on CPU) was not optimized and was the main reason why running mFAR was slow. Furthermore, future work in pruning may also further reduce the amount of training time needed.

---

> > ### Comment · Reviewer_Hpub · 2024-11-23
> >
> > Thanks for the authors' response. Empirically, it requires some experimentation to tune the weights for BM25F, and using 1 for each field is definitely the best option. An alternative approach is to use the average field score of G(*) derived from all queries in the training set.

---

> > ### Comment · Reviewer_xHpK · 2024-11-24
> > **Constant fields weights isn't a good idea**
> >
> > After reading the review of Hpub03 and the rebuttal:
> > 1. BM25F scores are surprisingly low, so it's worth rechecking. Perhaps, it's an issue with selecting constant weights and some fields, e.g., title are not very representative, so the results are skewed.
> > 2. But more importantly, for a truly good performance field weights need tuning. One powerful algorithm to do so is coordinate ascent and there's even a Python version so one doesn't have to mess with the Java library RankLib: https://github.com/jjfiv/fastrank
> > 3. A minor follow up: Which software do you use for BM25F implementation? Is it some hand-written solution or do you use a mature retrieval system like Elastic, Vespa, or maybe Pyserini?

---

> > > ### Author Response · Authors · 2024-11-24
> > > **Response to reviewers Hpub and xHpK**
> > >
> > > This response contains responses to both Hpub and xHpK. The main question we have for both reviewers: we will spend more time to include a better-tuned baseline for the paper, but given the remaining time in the rebuttal period and that our BM25F runs are fairly slow (30mins for an iteration of the fastest dataset, Prime), what would the reviewers find most useful for us to try during the next ~2 days? Note that the datasets have 5 (Mag), 8 (Amazon), and 22 (Prime) fields, respectively, so e.g. line search on Prime might be intractable.
> > >
> > > __To Hpub:__
> > >
> > > > Thanks for the authors' response. Empirically, it requires some experimentation to tune the weights for BM25F, and using 1 for each field is definitely the best option. An alternative approach is to use the average field score of G(*) derived from all queries in the training set.
> > >
> > > Thanks for the suggestion, and we can experiment with something like this as another BM25F as we update our paper. Using G(*) would require training a system like mFAR in the first place and so it might only be useful as guidance for good field values. A more interesting approach would be to learn it end-to-end as part of mFAR.
> > >
> > > > Thanks for authors response and additional experimental results in a such short period of time. Table retrieval is just one example of the multi-field ranking tasks extensively studied by the information retrieval community. Regarding datasets, there are several public datasets used for table-based QA [1-2], where table retrieval is a crucial step. In general, there are prior methods addressing multi-field ranking/learning, but none of them have been discussed or compared in this context [3-5].
> > >
> > > We will include a section of related works about the comparisons (and similarities) between table retrieval and (semi-)structured retrieval, including those you have mentioned. We agree it is a good idea to include [3-5] (we already have [3]). We will include a longer discussion on generally multi-field learning and why our setting differs.
> > >
> > > __To xHpK:__
> > >
> > > > BM25F scores are surprisingly low, so it's worth rechecking. Perhaps, it's an issue with selecting constant weights and some fields, e.g., title are not very representative, so the results are skewed.
> > >
> > > We agree that the scores are low, and this (like you mentioned) is indicative of assuming a uniform weight distribution for all fields, especially since uniform worked relatively better for MAG (5 fields) than the other two (8 and 22 fields).
> > >
> > > > But more importantly, for a truly good performance field weights need tuning. One powerful algorithm to do so is coordinate ascent and there's even a Python version so one doesn't have to mess with the Java library RankLib: https://github.com/jjfiv/fastrank
> > >
> > > Thanks for the suggestion, we can look into some coordinate ascent method for tuning the weights better, related to the main question above to both reviewers.
> > >
> > > > A minor follow up: Which software do you use for BM25F implementation? Is it some hand-written solution or do you use a mature retrieval system like Elastic, Vespa, or maybe Pyserini?
> > >
> > > We implemented the BM25F starting with [1] because it was most promising for future integration into our code and was the simplest starting point. We couldn’t find off-the-shelf Python implementations of BM25F. Elastic does not seem to have it, but possibly has considered it before [2]. Vespa does not seem to support it either. For BM25 (not “F”) baselines in the paper, we had used Pyserini and [bm25s](https://github.com/xhluca/bm25s), but both would have been harder to modify into BM25F than [1].
> > >
> > > [1] https://github.com/jxmorris12/bm25_pt
> > >
> > > [2] https://github.com/elastic/elasticsearch/issues/9609

---

> > > > ### Comment · Reviewer_xHpK · 2024-11-24
> > > > **to clarify on multi-field BM25**
> > > >
> > > > Sorry about the confusion. The original BM25F implementation is, indeed, may not be available. However, one could use weighted BM25 fields with weights learned using coordinate ascent. https://www.elastic.co/guide/en/app-search/current/relevance-tuning-guide.html
> > > >
> > > > Moreover, given the limited time for a rebuttal, it is not recommended (or even prohibited?) to run extensive additional experiments. I tend to accept the paper even in the absence of a multi-field BM25 baseline, but I will have to point this out in the final review.

---

> > > > ### Comment · Reviewer_Hpub · 2024-11-24
> > > >
> > > > - Regarding the use of G(*) for empirically selecting weights for BM25F, I would accept using the derived scores from your mFAR on the training set. If BM25F shows a significant improvement and outperforms BM25, it would strongly indicate that your mFAR is learning the correct weights.
> > > > - I understand that implementing BM25F yourself during the rebuttal process can be challenging. That's why I recommend the alternative approach of using the above method instead, to avoid the complexity of greedy-searching the weights.
> > > > - Another potential baseline to consider is the mixture of language models (i.e., eq 5 in [1]) which is also a common baseline used in table retrieval.
> > > >
> > > > [1] Combining Document Representations for Known-Item Search, SIGIR 2003

---

> > > > > ### Comment · Reviewer_xHpK · 2024-11-24
> > > > > **let's use proper weights for BM25**
> > > > >
> > > > > >Regarding the use of G(*) for empirically selecting weights for BM25F, I would accept using the derived scores from your mFAR on the training set.
> > > > >
> > > > > A word of caution here from a fellow reviewer: The scores that are good for a neural model aren't necessarily good for BM25.

---

> > > > > > ### Author Response · Authors · 2024-11-24
> > > > > > **Response to reviewer xHpK**
> > > > > >
> > > > > > __To reviewer xHpK__:
> > > > > >
> > > > > > > Sorry about the confusion. The original BM25F implementation is, indeed, may not be available. However, one could use weighted BM25 fields with weights learned using coordinate ascent. https://www.elastic.co/guide/en/app-search/current/relevance-tuning-guide.html
> > > > > >
> > > > > > Thank you for the clarification. The analogous baseline to your suggestion of BM25 with learned weights is our version of mFAR_{lexical} which contains BM25 for each field, weighted and learned through query conditioning. So, mFAR_{lexical} is our strongest learned lexical baseline.
> > > > > >
> > > > > > We are interested in your suggestion with RankLib for BM25F (even though you intended it for BM25). So we will look into it for learning the weights as a baseline for the paper, and so we might have results on a single dataset in the next couple of days.

---

> > > > > > > ### Comment · Reviewer_xHpK · 2024-11-24
> > > > > > > **mFAR_{lexical}  is worse than BM25? something is probably wrong**
> > > > > > >
> > > > > > > Hi, how is mFAR_{lexical} exactly produced? I looked through paper quickly, but didn't find a detailed explanation. How were the weights learned? If one implements a multi-field baseline correctly, it's at least not worse than BM25 for a combined field.
> > > > > > >
> > > > > > > So here's the question. Are you considering the following baseline:
> > > > > > > 1. Index and retrieve using the combined BM25-field using the BM25 for this field.
> > > > > > > 2. Re-rank top-K records using a weighted combination of all fields **including** the combined field.

---

> > > > > > > > ### Author Response · Authors · 2024-12-01
> > > > > > > > **Response to reviewers xHpK and Hpub**
> > > > > > > >
> > > > > > > > Thank you both, xHpK and Hpub, for all of the feedback. **Following these suggestions, we will include a more detailed section discussing the similarities and differences of our framework to BM25F, and especially about the limitations of applying BM25F to more than a few fields.** Notably, BM25F shares a similar motivation of decomposing fields and assigning (or learning) weights per field. We’ve realized that the actual implementation is considerably different because the entire corpus needs to be re-indexed for each new set of weights. This leads to two limitations of BM25F:
> > > > > > > >
> > > > > > > > 1) Coordinate ascent/grid-search/end-to-end based algorithms are expensive and would not be able to leverage the full scale of our datasets. For instance, to be able to run coordinate ascent, each iteration must re-index the corpus (* number of queries * number of fields) to obtain scores for each feature (field). This would nearly be intractable as corpus size scales and must be pre-computed or we must limit to a subset of the corpus/query set. Note that this could be mitigated if we consider using BM25F as a re-ranker only or only run coordinate ascent/training on a subset of the corpus and query set, but that was not the goal of our work.
> > > > > > > >
> > > > > > > > 2) Without an efficient trick (like negative sampling) applied and verified for BM25F, query-conditioning is also not tractable during training. This is even more of an issue at inference: for each test query $q$, we would predict a set of field weights that would require indexing of the full corpus (O(100k) docs * number of fields).
> > > > > > > >
> > > > > > > > We started implementation for a G(*) experiment that reviewer Hpub suggested, but ultimately did not complete during this period because of mainly limitation 2):
> > > > > > > > The main reason is that the cost was prohibitive: each query would take considerable time due to reindexing.
> > > > > > > > We also do not understand how such an experiment (or result) would fit within our paper. It does not qualify as a baseline since it requires training mFAR first. We would also not recommend running it in practice, as our method is scalable to any number of fields, but BM25F requires as many as num_fields * 2 + 1 independent grid searches [1], which we would have to optimize separately. It could be an analysis subsection, but we too share similar skepticism about whether neural weights are meaningful in the first place. When we looked at these weights in earlier analysis, we found them difficult to interpret.
> > > > > > > >
> > > > > > > > These limitations of BM25F are challenges that go beyond our work. We will choose a reasonable setup to compare against, and discuss these limitations in an updated section in related work.
> > > > > > > >
> > > > > > > > > So here's the question. Are you considering the following baseline: Index and retrieve using the combined BM25-field using the BM25 for this field. Re-rank top-K records using a weighted combination of all fields including the combined field.
> > > > > > > >
> > > > > > > > Thank you for this suggestion, we did overlook this and we did not report results on the paper yet; our mFAR_{lexical} model does not consider the combined field. We agree that it should use the combined field (and so it should not be worse than BM25). We will update these numbers with the following setup that we ran following your suggestion.
> > > > > > > >
> > > > > > > > In step 1., we instead retrieve candidates based on both the combined field and the individual subfields. We obtained the following scores (Amazon crashed), which continues to support that lexical signal is effective on MAG (competitive with our best models) but not on Prime, and these scores are better than both the mFAR_{lexical} model and when we used BM25 on just the single (combined) field.
> > > > > > > >
> > > > > > > > | | H@1 | R@20 | MRR |
> > > > > > > > | -------- | ------- | ------- | -------- |
> > > > > > > > | MAG |  0.518 | 0.712 | 0.605 |
> > > > > > > > | Prime   |   0.259 | 0.507 | 0.352 |
> > > > > > > >
> > > > > > > > Note this is not an “external baseline” of a multi-field model because still we apply our own query-conditioning adaptation, and so it should be viewed more as a lexical-only version of mFAR. We will also report a version without query-conditioning to obtain a multi-field lexical baseline.
> > > > > > > >
> > > > > > > > [1] Microsoft Cambridge at TREC–13: Web and HARD tracks, https://trec.nist.gov/pubs/trec13/papers/microsoft-cambridge.web.hard.pdf, 2004

---

> > > > > > > > > ### Comment · Reviewer_Hpub · 2024-12-02
> > > > > > > > >
> > > > > > > > > Thank you for your hard work during the rebuttal phase.
> > > > > > > > >
> > > > > > > > > I believe reviewer xHpK is asking for two baselines based on BM25:
> > > > > > > > >
> > > > > > > > > - One using a combined field
> > > > > > > > > - Another with weighted versions of BM25
> > > > > > > > > Additionally, I think it would also be valuable to include a dense version of both baselines, utilizing Contriever.
> > > > > > > > >
> > > > > > > > > While I appreciate the method proposed, my main concern is the absence of these additional baselines and the lack of further discussion on the results. From Table 2, it’s clear that the lexical-based retriever performs well on the Amazon dataset, while the dense retriever outperforms on the Prime dataset. Notably, MFAR2, which uses a single field, achieves the best performance on Amazon. I would expect that a weighted BM25 using a single field would yield much higher performance than the BM25 reported here. Similarly, for the Prime dataset, I anticipate that the weighted version of Contriever with a single field would outperform the reported results. These comparisons are crucial for understanding the contexts in which each method excels. Additionally, reporting the learned importance weights for different fields and scorers would provide more valuable insights into model behavior.

---

> > > > > > > > > > ### Author Response · Authors · 2024-12-02
> > > > > > > > > >
> > > > > > > > > > Hi, we interpreted xHpK as asking for one additional experiment, and perhaps they can clarify.
> > > > > > > > > >
> > > > > > > > > > The experiment using only a single  “combined field”, i.e. to treat the doc as a single sequence, is already represented by the “BM25” row in Table 2. Analogously, the dense versions are represented by “Contriever-FT” and mFAR_{dense}. However, we agree that we also need to run the analogous dense version with combined+separate fields as we did for reviewer xHpK in the lexical setting. In this setting, we **include** query conditioning. We will add these to the table along with some discussion around these results once we have obtained all of them, if the results are interesting.
> > > > > > > > > >
> > > > > > > > > > Once again, we want to emphasize to all readers, that these experiments are **not baselines** (and we will make this clear in the paper too). All of our weighted models combine and jointly learn query-dependent weights.
> > > > > > > > > >
> > > > > > > > > > A baseline (/ablation) is to run all of these models **without** query conditioning, i.e. all the field weights are static for all queries, which might be what you implied. We will also run these experiments without the query-conditioning, like we have already done for some experiments in the paper, and add these as baselines. As we have seen, query-conditioning is quite important, and so we expect most/all of these baselines (without query conditioning) to perform no better than their more expressive counterpart which is allowed to condition on the predicted weights.
> > > > > > > > > >
> > > > > > > > > > After running the experiments for the paper, we will discuss the results if they are different from what we have already discussed. The discrepancies between which method works best on each dataset is something we already touch on a little bit within the paper, but one takeaway (which you listed originally as S3) is that no method is best on all datasets, and that appears to continue to be the case with the stronger mFAR lexical combination that xHpK proposed.
> > > > > > > > > >
> > > > > > > > > > > I would expect that a weighted BM25 using a single field would yield much higher performance than the BM25 reported here….
> > > > > > > > > > >  Similarly, for the Prime dataset, I anticipate that the weighted version of Contriever with a single field would outperform the reported results.
> > > > > > > > > >
> > > > > > > > > > Could you clarify what you mean by “weighted BM25 using a single field” (and same for Contriever)? We (authors) have conflicting interpretations of this phrase. Specifically, if the fields in the doc are A, B, C, are you suggesting $S(q, [A;B;C])$ (where the fields are concatenated into a single doc), or $\lambda_A S(q, A) + \lambda_B S(q, B) + \lambda_C S(q, C) + \lambda_D S(q, [A;B;C])$, where the $\lambda$s are learned, or something else? Here, $S$ is any scoring function like BM25 or Contriever similarity.

---

> > > > > > > > > > > ### Comment · Reviewer_Hpub · 2024-12-02
> > > > > > > > > > >
> > > > > > > > > > > Sorry for the confusion. I mean the latter equation (without the last term) and all different lambda-s are just hyperparameters.

---

> ### Comment · Reviewer_Hpub · 2024-11-23
>
> Thanks for authors response and additional experimental results in a such short period of time. Table retrieval is just one example of the multi-field ranking tasks extensively studied by the information retrieval community. Regarding datasets, there are several public datasets used for table-based QA [1-2], where table retrieval is a crucial step. In general, there are prior methods addressing multi-field ranking/learning, but none of them have been discussed or compared in this context [3-5].
>
>
>
> [1] Open Question Answering over Tables and Text, ICLR 2021
>
> [2] ConvFinQA: Exploring the Chain of Numerical Reasoning in Conversational Finance Question Answering. EMNLP 2022
>
> [3] Neural Ranking Models with Multiple Document Fields, WSDM 2018
>
> [4] GLOW: Global Weighted Self-Attention Network for Web Search, BigData 2018
>
> [5] Multi-field Learning for Email Spam Filtering, SIGIR 2010

---

> ### Comment · Reviewer_xHpK · 2024-12-02
> **a nice-to-have (but not must-have) experiment clarification**
>
> Yes, I asked for one additional, but **optional** experiment where BM25 scores are learned using coordinate ascent. Regarding the **combined field** ask: What I suggest for this experiment is to have a combined field as one more field in the mix. In fact, one can experiment with inclusion/exclusion of this field from the mix.
>
> I would like to emphasize that I consider this experiment as a "nice-to-have" rather than "must-have".
>
> PS: the rationale for adding a combined field to the mix: With enough training data, the results will never be **worse** than using only a single combined field. Why? Because, the learning algorithm can assign weight zero for any other field except the combined field. This would permit diagnosing issues with the learning algorithm and what not. However, in my experience with multi-field ranking is that additional fields usually improve outcomes at least a bit. Check out, e.g., an MS MARCO **document ranking** leaderboard, which has both single-field and multi-field BM25 runs.

---

> ### Comment · Reviewer_Hpub · 2024-12-02
>
> Based on the discussions, I have decided to slightly increase the score.
>
> I encourage the authors to consider adding additional baselines, such as BM25 or Contriever, that integrate scores from multiple fields with weights as hyperparameters (I am very interested in the correlation between the tuned weights and the learned weights produced by your method). This could further strengthen the study.
>
> Once again, I thank the authors for their active engagement and for conducting additional experiments during the rebuttal phase.

---

> > ### Author Response · Authors · 2024-12-02
> >
> > Thanks for the clarification on the equation, that's our mFAR_lexical model (and mFAR_dense) model except the ones we report in the paper use query conditioning. Removing the query conditioning from these models is a baseline that we overlooked but will certainly add in (also for dense) as we round out all of the experiments for completeness.
> >
> > Finally, thank you both again for your patience and invaluable feedback over the last several days.

---

### Official Review · Reviewer_vQrJ · 2024-11-03

**Soundness:** 3
**Presentation:** 3
**Contribution:** 3
**Rating:** 8
**Confidence:** 4

**Summary:**

This paper presents Multi-Field Adaptive Retrieval (MFAR), a framework for retrieving structured documents by dynamically weighting document fields based on query relevance. By combining lexical and dense retrieval methods, MFAR improves retrieval accuracy on the STaRK dataset, surpassing state-of-the-art baselines and highlighting the potential of adaptive, hybrid approaches for structured data retrieval.

**Strengths:**

1. The paper proposes a new approach that incorporates structured document retrieval, where documents contain multi-field information. The proposed method adaptively controls the information by using a weight adjustment strategy based on the query.
2. The experimental results demonstrate that mFAR framework over various baselines on the STaRK dataset.
3. The authors present detailed analysis demonstrating why their hybrid approach can outperform baselines through the experiments of both multi-field vs. single-field and hybrid vs. lexical/semantic similarity.
4. The paper is well-structured, motivated, and written, thus easy to follow.

**Weaknesses:**

1. Though several recent retrievers [1] and [2] have been proposed, Contriever is used as a representative of dense retrievers without sufficient discussion. I am concerned that the conclusion may change if the authors use the recent retrievers. Especially, since FollowIR is instruction-tuned, it may be more robust against negation in the query, which the authors pointed out as a possible issue of the single-field method.
2. The adaptive weighting mechanism in MFAR relies on training within specific domains and structured datasets, making it potentially sensitive to domain-specific data characteristics. This might lead to suboptimal field selection or scoring when the document structure is inconsistent across the corpus.

[1] Weller, Orion, et al. "FollowIR: Evaluating and Teaching Information Retrieval Models to Follow Instructions." arXiv preprint arXiv:2403.15246 (2024).

[2] Xiao, Shitao, et al. "C-pack: Packaged resources to advance general chinese embedding." *arXiv preprint arXiv:2309.07597* (2023).

**Questions:**

Regarding Weakness 1 in the previous section, did you try to use other dense retrievers and do you have any insights about experiments with better retrievers?

---

> ### Author Response · Authors · 2024-11-19
> **Response to reviewer**
>
> We thank you for your concerns, as it helps us clarify the design decisions for our framework.
>
> > Though several recent retrievers [1] and [2] have been proposed, Contriever is used as a representative of dense retrievers without sufficient discussion. I am concerned that the conclusion may change if the authors use the recent retrievers. Especially, since FollowIR is instruction-tuned, it may be more robust against negation in the query, which the authors pointed out as a possible issue of the single-field method.
> > Regarding Weakness 1 in the previous section, did you try to use other dense retrievers and do you have any insights about experiments with better retrievers?
>
> We had tried GTR-T5 [1] initially and found that Contriever-MSMARCO [2] was performing significantly better in initial experiments for the single-field finetuning baselines, across multiple hyperparameters. In [6], they also report multiple models of various sizes, like roberta, ColBERTv2 [7], GritLM (instruction-tuned, 7B-size) [3], LLM2Vec [4], and ada-002 [5], many of these models recent and the larger models are not fine-tuned. Yet the scores we obtained by fine-tuning Contriever-MSMARCO (110M params) (in Table 2, Contriever-FT) is comparable or surpasses most of their baselines. Thus, we believe that Contriever-MSMARCO is a strong starting point to demonstrate modeling contributions, especially as we are interested in fully fine-tuning the model for end-to-end training, and that many of the other models are too large.
>
> We did not see promising preliminary results without the ability to fine-tune the encoder, and so we did not try directly applying mFAR to the frozen LLMs. Given more compute capacity, we expect that fine-tuning any of the larger competitive models (like GritLM-7B or ada-002) could yield higher-scoring baselines, and further applying mFAR on top of that with fine-tuning could lead to similar gains/trends.
>
> We did not consider treating instruction fine-tuned methods, like FollowIR, differently, as our work was focused on document decomposition and that most of the queries in STaRK are relatively straightforward (additionally, as noted above, GritLM is an instruction-tuned model that performs similarly to other models). Your comment raises a valid point that there are other solutions on the encoder-side to robustness – negation specifically. We do not think the point invalidates our method, which demonstrates a (separate, unintended) solution to negation on the document-side. However, we thank you for the suggestion, and we will make it clearer that there are other solutions, like FollowIR/GritLM, and that the figure presented in our error analysis is not meant to represent an explicit goal of solving negation/robustness.
>
> > The adaptive weighting mechanism in MFAR relies on training within specific domains and structured datasets, making it potentially sensitive to domain-specific data characteristics. This might lead to suboptimal field selection or scoring when the document structure is inconsistent across the corpus.
>
> We do agree that training is domain-specific. However, this is also the case for any single-field fine-tuning setup where a retriever must be finetuned on the dataset for best performance.
>
> Regarding inconsistent structure: entries in the Prime dataset are inconsistent. Only some entities (genes/proteins) have “interacts_with” field, while others (drugs) do not have that but instead have “side_effects.” No entity has every field. To address this, we artificially zero out nonexistent fields for each sample during training, which still allows the model to learn with all fields, even though no document contains all possible fields.
>
>
> [1] Large Dual Encoders Are Generalizable Retrievers, https://arxiv.org/abs/2112.07899, 2021
>
> [2] Unsupervised Dense Information Retrieval with Contrastive Learning, https://arxiv.org/abs/2112.09118, 2021
>
> [3] Generative Representational Instruction Tuning, https://arxiv.org/abs/2402.09906, 2024
>
> [4] LLM2Vec: Large Language Models Are Secretly Powerful Text Encoders, https://arxiv.org/abs/2404.05961, 2024
>
> [5] ada-002, https://openai.com/index/new-and-improved-embedding-model/
>
> [6] STaRK: Benchmarking LLM Retrieval on Textual and Relational Knowledge Bases, https://arxiv.org/abs/2404.13207, 2024
>
> [7] ColBERTv2: Effective and Efficient Retrieval via Lightweight Late Interaction, https://arxiv.org/abs/2112.01488, 2021

---

> > ### Comment · Reviewer_vQrJ · 2024-11-23
> >
> > Thank you for the detailed responses that clarified my concerns. Especially, the discussion about retrievers was insightful and provided a solid rationale for using Contriever-MSMARCO as a starting point. I have raised my score from 6 to 8.

---

### Official Review · Reviewer_xHpK · 2024-11-04

**Soundness:** 4
**Presentation:** 4
**Contribution:** 3
**Rating:** 8
**Confidence:** 4

**Summary:**

This is a timely "revisit" of a multi-field retrieval problem with a proposal of multi-field adaptive retrieval that combines sparse-lexical and learned-dense-vector-similarity scores using field-weights, which are learnable and adaptive. Namely, each weight is a neural network incorporating a query representation and a field-specified learned weight vector.

The paper is very nicely written, it has a convincing set of results, as well as thorough literature review.

**Strengths:**

* The paper revisits an important problem.
* The paper is very well written
* Experimental results are convincing and have quite a few baselines.
* The method is simple and elegant yet effective

**Weaknesses:**

The only noticeable weakness is the need to compare to a better multi-field BM25 baseline, ideally, where field weights are learned using coordinate ascent (e.g., usling RankLib, or FastRank: https://github.com/jjfiv/fastrank). I do not think, however, that this should be a deal breaker. However, we encourage authors to list this as a limitation of their work.

**Questions:**

The following questions were fully answered/addressed by the authors:
1. What is exactly used for dense/vector retrieval?
2. How did you decide on selection of top-100 records for each field (L885)?
3. In Eq. 4, what is exactly q in G(q, f, m): I assume it should be a dense query encoder, but it's not clear from the text (at least for me).

---

> ### Author Response · Authors · 2024-11-19
> **Response to reviewer**
>
> We thank you for your questions, which will clarify our paper, and we are glad that you find our paper well written.
>
> > What is exactly used for dense/vector retrieval?
>
> > In Eq. 4, what is exactly q in G(q, f, m): I assume it should be a dense query encoder, but it's not clear from the text (at least for me).
>
> Thank you for catching these details. For dense/vector retrieval, we use an off-the-shelf embedder (Contriever-MSMARCO [1]) to encode the query $q$ to $\bf{q}$ and encode the text from fields $x_f$ to $\bf{x}_f$, decomposed from document $d$. For the dense retrieval, we can then compute similarity (unnormalized inner product) between $\bf{q}$ an $\bf{x}_f$. The same $\bf{q}$ is used in $G$.
>
> We will make this notation clearer, as presently we’ve left out the definition of $\bf{q}$.
>
> > How did you decide on the selection of top-100 records for each field (L885)?
>
> We followed the experimental setting of k=100 in the original Contriever paper and past work. To efficiently decide which documents belong in the top-k, we only fully-score documents if one of its fields belongs in the top-k for at least one of the $s_f^m$ functions. This subset of documents is sorted and the top-k (k=100) is used as the document ranking for evaluation.
>
> [1] Unsupervised Dense Information Retrieval with Contrastive Learning, https://arxiv.org/abs/2112.09118, 2021

---

> > ### Comment · Reviewer_xHpK · 2024-11-24
> > **ack**
> >
> > Thank you for clarifying. Turns out that you did mention using a finetuned version of the Contriever, but I missed it.
> >
> > More importantly, though, as another reviewer pointed out, one can compare against BM25F as well. Please, see my reply there.

---

### Official Review · Reviewer_JpGq · 2024-11-04

**Soundness:** 3
**Presentation:** 4
**Contribution:** 3
**Rating:** 8
**Confidence:** 4

**Summary:**

This paper exploits the structure in various IR corpora by learning query, corpus, field dependent weights across vector retrieval and token retrieval. The paper compares the methods to a number of strong baselines and analyzes a thorough set of ablation experiments to identify difference makers across different benchmarks. This paper is well written, easy to read and has a comprehensive set of references.

**Strengths:**

The paper is well motivated and well written. Many documents naturally have structure. Exploiting it should lead to better retrieval quality. However, doing so depends on the query, corpus and kind of retrieval. This paper comes up with an elegant and intuitive formulation to learn all of these weights during training. The baselines are well chosen, ablation experiments are extensive and the references are comprehensive.

**Weaknesses:**

The main weakness, and kudos to the authors for discussing this, is what they mention in Section 4 "Multi-field vs Single-field": "A side-by-side comparison of the single field models against their multi-field counterparts shows mixed results".  The difference in averages between mFAR_2 and mFAR_all doesn't appear statistically significant. The primary gains (looking at mFAR_2 vs mFAR_all) seem to be in the prime data set which has a lot of fields. Should the paper instead focus on adaptive hybrid retrieval, and choose hybrid retrieval baselines?

**Questions:**

1. Are "Dense only" in Table 4 and "mFAR_dense" in Table 2 the same (the numbers are close but different). Were mFAR_dense and mFAR_lexical trained separately or trained together and one component ablated?

2. See the weakness section, which I phrased as a question.

---

> ### Author Response · Authors · 2024-11-20
> **Response to review**
>
> We thank you for the very specific comments, and we are happy to hear that the motivation is clear to you and that the paper is easy to read. Below, we address what was mentioned:
>
> > The main weakness, and kudos to the authors for discussing this, is what they mention in Section 4 "Multi-field vs Single-field": "A side-by-side comparison of the single field models against their multi-field counterparts shows mixed results". The difference in averages between mFAR_2 and mFAR_all doesn't appear statistically significant. The primary gains (looking at mFAR_2 vs mFAR_all) seem to be in the prime data set which has a lot of fields. Should the paper instead focus on adaptive hybrid retrieval, and choose hybrid retrieval baselines?
>
> We believe the paper’s current focus (as stated in the abstract) is on both adaptive retrieval and hybrid retrieval, although we do lean into the adaptive part more, as even in the mFAR_2 run, we are conditioning on the query to determine how to weight the (single) dense score against the (single) BM25 score.
>
> We looked for and did not find hybrid retrieval baselines that could be directly compared to our setting. Some past works (which we will discuss more in the paper) [1, 2] typically find complementary features of dense and lexical scorers as part of a full retriever stack, which include re-ranking or alternating between lexical and dense components. Meanwhile all of our baselines are focused on obtaining a single score for document ranking. [3, 4] combines lexical and semantic features together into a single score, but their lexical features are also dense and trained end-to-end (not sparse like our BM25 scores). Due to these differences, we did not make a comparison within the paper.
>
> Furthermore, many of these papers are not easily reproduced or adaptable as they do not release their code. So, we thought of a simpler hybrid model baseline ablation that we can run that still uses only a semantic encoder and BM25. This is a mFAR_2 model without the adaptive query conditioning. Instead, it learns a single weight for the dense score and a single weight for the sparse score, to be used for all examples. In a preliminary run, it scored 0.550 for Hit@1 on Amazon, which is a little lower than the 0.574 that we report in the paper. This is early evidence both that a simple hybrid baseline is actually quite strong too – it alone would be state-of-the-art – and that our adaptive weighting can still provide benefits on top of that.
>
> We also agree that the “mixed results” confused us as they don’t lead to crisp conclusions about separating into multiple fields. To investigate further, we trained another variant of mFAR named mFAR_{1+n}. This one consists of a single sparse score and multiple dense scores (one per field) – we can view this as somewhere in between mFAR_{all} and mFAR_2. This model performs at or above the level of mFAR_2 across the 3 datasets, which suggests a conclusion that there is a benefit to decomposing the dense scorer into fields. We show the results of both models in the table below, and will also update our preprint with this new information.
>
> |     | Amazon - H@1 | R@20 | MRR | MAG - H @1 | R@20 | MRR | Prime - H@1 | R@20 | MRR |
> | -------- | ------- | ------- | -------- | ------- | ------- | -------- | ------- | ------- | ------- |
> | mFAR$_2$  | 0.574 | 0.663 | 0.681 | 0.503 | 0.721 | 0.603 | 0.227 | 0.495 | 0.327| 0.435| 0.626 | 0.537 |
> | mFAR$_{1+n}$ | 0.565 | 0.659 | 0.674 | 0.511 | 0.748 | 0.611 | 0.359 | 0.650 | 0.469 |
>
> > Are "Dense only" in Table 4 and "mFAR_dense" in Table 2 the same (the numbers are close but different). Were mFAR_dense and mFAR_lexical trained separately or trained together and one component ablated?
>
> In Table 2, mFAR_dense and mFAR_lexical are mFAR models trained with only dense scorers or lexical scorers, respectively. In Table 4, “Dense only” refers to a post-hoc masked version of mFAR_all, where all the weights associated with lexical scorers are set to 0, which would leave behind only dense scores.
>
> In other words, if $|F|$ is the number of fields in a dataset, mFAR_dense has $|F|$ weights, mFAR_lexical also has $|F|$ weights. Meanwhile, mFAR_all has 2$|F|$ weights (|{dense, lexical}| * $|F|$). Dense-only and Lexical-only makes inference-time adjustments to mFAR_all. All 2$|F|$ weights are predicted, but half of them are clamped, at inference time, to 0. We can clarify in Section 5.2.
>
> [1] Complementing Lexical Retrieval with Semantic Residual Embedding, https://arxiv.org/abs/2004.13969, 2020
>
> [2] On Complementarity Objectives for Hybrid Retrieval, https://aclanthology.org/2023.acl-long.746/, 2023
>
> [3] A Dense Representation Framework for Lexical and Semantic Matching, https://arxiv.org/abs/2206.09912, 2022
>
> [4] UnifieR: A Unified Retriever for Large-Scale Retrieval, https://arxiv.org/abs/2205.11194, 2022

---

> ### Comment · Reviewer_JpGq · 2024-11-27
>
> Thank you for the detailed and thoughtful response. In general, I agree that while learning adaptive retrieval across dense and sparse scorers, taking multiple fields into account seems reasonable. Perhaps future work will fully bear out the potential for multi-field retrieval. The time boundedness makes this hard, but I agree with other reviewers that it is rather important to compare against strong baselines for multi-field retrieval, some of which predate neural models. Perhaps the authors can acknowledge the difficulty in including some baselines like BM25F, referencing them in the Related Work and Future Work sections.
>
> Overall, I think the community will benefit from the publication of this work. I will raise my score to 8.

---

### Official Review · Reviewer_EjWS · 2024-11-05

**Soundness:** 3
**Presentation:** 3
**Contribution:** 3
**Rating:** 8
**Confidence:** 4

**Summary:**

The paper presents a framework to improve document retrieval (Multi-Field Adaptive Retrieval, MFAR). Whereas document retrieval typically uses unstructured data for tasks such as retrieval augmented generation, MFAR is designed for structured data where documents have multiple fields (e.g., titles, abstracts, author information). MFAR first decomposes documents into individual fields and then learns a model that adaptively weights fields based on the input query. The framework uses both dense and lexical methods for retrieval, optimizing the combination of these representations per field to improve retrieval performance. Experimens show that MFAR outperforms previous retrieval models in complex document retrieval on the STaRK dataset.

**Strengths:**

Rereach on structured document retrieval is highly relevant, especially for RAG approaches. The retrieval is well designed, using a hybrid and adpative query scoring mechanism, using both dense and lexical methods as well as a ranking strategy. The evaluation is thorough, and the paper is well-structured and generally well-written.

**Weaknesses:**

The fine-tuning approach makes the approach specific to a set of fields from a dataset. Information overlap in fields (see lines 416-424) might intrudice some redundancy to the retrieval process.

**Questions:**

How much robust is the framework to variations in the number of fields, e.g., regarding field information overlap?

---

> ### Author Response · Authors · 2024-11-20
> **Response to review**
>
> We thank you for your comments and appreciation of our approach. We are glad that you find the paper well-written and well-structured. Below, we address the weakness(es) and question that you brought to our attention:
>
> > The fine-tuning approach makes the approach specific to a set of fields from a dataset.
>
> We agree with this. However, this still gives us a lot of flexibility. The main appeal of our modeling approach with a specific set of fields denoted is the fact that finetuning can be personalized for a specific dataset. If we do not know what queries will ask about, we can be more inclusive and include everything during fine-tuning (as computation budget allows), and our framework will eventually learn the important fields; the unimportant fields would anyways be assigned lower weights. As a future extension, we could even automatically learn to prune the unimportant fields either post-hoc or during training.
>
> > Information overlap in fields (see lines 416-424) might intrudice some redundancy to the retrieval process.
>
> We do agree that there is redundancy with the tokens and phrasings that might be retrieved, and we do not claim that fields are completely orthogonal. In fact, the redundancy is warranted as fields that highlight the same proper noun or phrasing should be promoted as our scoring (especially with a lexical scorer, such as BM25) benefits from repeated instances of a specific token or repetitions in different contexts.
>
> > How much robust is the framework to variations in the number of fields, e.g., regarding field information overlap?
>
> This is an interesting question. In some preliminary ablations, we attempted to train a model without certain fields. Specifically for MAG, we tried experiments with only 1, 2, or 3 of the 5 fields. We found that the score would monotonically increase as we increased the number of fields, and that all the fields were needed for best performance.
>
> However, this might not be the case for all datasets. As we can see from the full results of masking out fields (Sec 5.3; Appendix D) for Amazon, some fields can be entirely removed from a fully-trained model without affecting the score, which suggests there is some degree of robustness. In particular, there are cases (like “qa” for Amazon, Table 5) where masking out either the lexical or dense scorer results in no drop, but masking out both results in a substantial drop in performance. This suggests that the model can be robust to redundant information being removed.

---

> > ### Comment · Reviewer_EjWS · 2024-11-25
> >
> > Thank you for the response. I will keep my original score.

---

### Official Review · Reviewer_ZnRT · 2024-11-06

**Soundness:** 3
**Presentation:** 3
**Contribution:** 2
**Rating:** 5
**Confidence:** 3

**Summary:**

The paper is concerned with the retrieval of documents which are multi-field, i.e, composed of multiple attributes such as title, authors and content. It proposes a method called MFAR which combines the use of different scoring methods, both lexical (word-based) and dense (embedding-based), for each field, allowing the model to adaptively predict the importance of each field for a given query.

The authors conduct experiments on three datasets (product reviews, scientific papers and biomedical studies) that demonstrate that MFAR outperforms existing retrieval methods, achieving state-of-the-art results in structured data information retrieval. The study explores the benefits of the multi-field approach and the hybrid use of scoring methods to improve retrieval accuracy, showing that, instead of choosing between dense or lexical-based scorers alone, one can combine them in a hybrid fashion. An adaptive weighting technique is provided to combine those scores given a query.

**Strengths:**

The paper tackles a relevant problem, although the proposal has some limitations discussed below. The paper is well written and easy to understand. Originality is rather limited, since the semi-structured retrieval has been researched for a long time. The significance of the results is also limited because the approach is rather simple, and the experimental evaluation focuses on a recently published benchmark. The paper quality is good to its purposes, despite some adhoc design decisions.

**Weaknesses:**

It is not clear to me whether the fields may be considered indepent, so that the summation of the field-related scores suffices for determining the overall score. That is, it seems intuitive that there are correlations among the field instances and they may bias the result, as was extensively researched in the information retrieval area.

Another field-related issue is regarding the process of selecting the fields that will be considered in the whole process.

Overall, the proposal is simple and basically consists of combining scoring functions associated with fields, without considering their correlations and other characteristics that may either characterize the task or explain problems or failures. For instance, although the paper focuses on the information carried by the fields, it seems intuitive to mix the aggregated value of the fields with the remaining text, exploiting eventual information there.

The experimental result also needs to be improved, as detailed next. First of all, the two experimental hypothesis seem to be too simple, thus quite easy to demonstrate. The advantages of using document structure are expected, in particular considering the additional information given to the models. The expected gains of hybrid approaches are also quite predictable. In both cases, it would be interesting to somehow derive upper bounds on the gains, so that the results go beyond benchmark-based evidence.

On the other hand, it is also necessary to clear outline the limitations of such hybrid approaches and how they may be addressed. I really missed confidence intervals or similar statistical significance metrics on the results. For instance, on Table 2, the results may be too close and, despite the bold highlight, it is not clear how far the results are from the remaining of the baselines.

One terminology issue is regarding the difference between structured and semi-structured and where the paper fits. It seems to me that semi-structured is the proper jargon.

**Questions:**

Are the scoring dimensions really orthogonal, enabling summation as the summarization metric?

Aren´t there additional baselines and data sets that may be used in the experiments?

---

> ### Author Response · Authors · 2024-11-20
> **Response to review, part 1**
>
> We appreciate your comments on our paper. We are glad that you find our paper easy to read and the quality good. Here, we lay out some thoughts regarding your comments:
>
> > It is not clear to me whether the fields may be considered indepent, so that the summation of the field-related scores suffices for determining the overall score. That is, it seems intuitive that there are correlations among the field instances and they may bias the result, as was extensively researched in the information retrieval area.
>
> > Q1: Are the scoring dimensions really orthogonal, enabling summation as the summarization metric?
>
> In our setup, we do not guarantee a notion of independence for the fields. We use different scorers for each field, so for models that use both lexical and dense scorers, both scorers score the same underlying text, resulting in likely dependence.
>
> We would be happy to include the references of needing independence if you could give us examples of these references showing that correlations may bias the result. Do note that we do achieve good performance with the assumption that there are potential dependencies between fields. This result suggests that the model is appropriately weighting the fields, even though there may be overlapping information in the field texts. We will add this discussion regarding independent and dependent fields to the paper (as EjWS also brought it up).
>
> > Another field-related issue is regarding the process of selecting the fields that will be considered in the whole process.
>
> Our goal is to provide a flexible framework to automatically select any and all fields that are most salient to an individual use case. Our query-conditioning approach allows the on-the-fly weighting of fields, lowering the weight for fields that are deemed less useful and increasing the weight for those fields that are more relevant to the query. This way, we don’t need to pre-select specific fields (we can use all of them in the dataset). This is relatively lightweight, adding a small number of parameters to the model (only 768 per field) which is minor compared to the 110M-size of the full model.
>
> > Overall, the proposal is simple and basically consists of combining scoring functions associated with fields, without considering their correlations and other characteristics that may either characterize the task or explain problems or failures. For instance, although the paper focuses on the information carried by the fields, it seems intuitive to mix the aggregated value of the fields with the remaining text, exploiting eventual information there.
>
> Our framework aims to do retrieval with a generalizable approach. Thus, finding correlations for each dataset between each field may be impractical. As for mixing with the remaining text, given a single field, we are including information under that field – for instance, if our fields of interest are “Book Title”, “Author”, and “Book Text”, we are including the texts “Moby Dick”, “Herman Melville”, and “Call me Ishmael. Some years ago… [truncated at context size]” respectively. So, we are mixing the aggregated value of the field with the full text already, to our understanding.
> We apologize if we have misunderstood your comment, and it would be illustrative if you could give an example or clarify what we do not already mix into the summation.
>
> > The experimental result also needs to be improved, as detailed next. First of all, the two experimental hypothesis seem to be too simple, thus quite easy to demonstrate. The advantages of using document structure are expected, in particular considering the additional information given to the models. The expected gains of hybrid approaches are also quite predictable. In both cases, it would be interesting to somehow derive upper bounds on the gains, so that the results go beyond benchmark-based evidence.
>
> For many research problems, simple solutions are preferred over complex solutions, so we do not believe simple solutions and experimental designs (in our case, query conditioning) are a weakness. For example, zero-shot chain-of-thought [2] is also a simple method with immense impact for LLMs. Though it may seem intuitive that keeping document structure improves performance, we show that in settings where we do not utilize query conditioning, performance is far worse than simply training a retriever on entire documents. Thus, it is not trivial to take  advantage of document structure.
>
> Furthermore, it is also not necessarily obvious that hybrid approaches will always produce better results. Despite Amazon obtaining relatively high performance for lexical-only and dense-only scorers, we find that the combination does not always achieve the best performance (in the case of mFAR with multiple fields).
>
> On upper bound: could you clarify what type of experiment you would like to see? Or are you looking for something theoretical, and can you explain what specifically?

---

> ### Author Response · Authors · 2024-11-20
> **Response to review, part 2**
>
> > On the other hand, it is also necessary to clear outline the limitations of such hybrid approaches and how they may be addressed. I really missed confidence intervals or similar statistical significance metrics on the results. For instance, on Table 2, the results may be too close and, despite the bold highlight, it is not clear how far the results are from the remaining of the baselines.
>
> Thank you for the excellent suggestion. As each (model, dataset) configuration requires training a full model on multiple GPUs in parallel, we will not have significant metrics completed in time during this rebuttal period. However, we will run each of the main mFAR models across multiple random seeds and compute confidence intervals for each model and will update the preprint.
>
> Anecdotally, we saw fluctuations of as much as 0.01-0.02 for each metric, but we agree this is not a substitute for more rigorous testing across all (model, dataset) combinations.
>
> > Q2: Aren´t there additional baselines and data sets that may be used in the experiments?
>
> Of course, there are always additional models and datasets. Could you be specific about which dataset or model you think would improve the paper?
>
> We already report several baselines, and [1] reports even more in their study on this dataset. Reviewer Hpub has brought up table retrieval as a task, which we also describe in related work. We will work on that for the final version, but may not complete them in time during this rebuttal period. Note that not all datasets are suitable for multi-field retrieval – standard datasets like MSMARCO or NQ do not have semi-structured documents, and so we would not expect mFAR to perform well on them (nor do we claim it would).
>
> > One terminology issue is regarding the difference between structured and semi-structured and where the paper fits. It seems to me that semi-structured is the proper jargon.
>
> Thank you for this suggestion; we will change the terminology to “semi-structured.”
>
>
> [1] STaRK: Benchmarking LLM Retrieval on Textual and Relational Knowledge Bases, 2024, https://arxiv.org/abs/2404.13207
>
> [2] Large Language Models are Zero-Shot Reasoners, 2022

---

> > ### Author Response · Authors · 2024-12-01
> > **Addressing your concerns**
> >
> > Hi reviewer ZnRT, did we address your concerns? We would like the opportunity to do so if not.

---

### Meta-Review · Area_Chair_XvjQ · 2024-12-12

**Metareview:**

The authors propose a framework for document retrieval for structured documents that may have multiple fields like title, abstract, etc. The contribution works on individual fields and adaptively aggregates the results so that different queries lead to different weightings of the fields. The paper is interesting and addresses an important problem, is well written and proposes an intuitive and rather straight forward but elegant technique that also yields convincing empirical results. The contribution was well received by the reviewers and should be of interest to the community as well.

**Additional Comments On Reviewer Discussion:**

There was quite an exchange between author(s) and reviewers during the rebuttal.

---

### Decision · Program_Chairs · 2025-01-22

Accept (Spotlight)